Manuscript prepared for Atmos. Chem. Phys.
with version 2015/04/24 7.83 Copernicus papers of the LaTeX class copernicus.cls.
Date: 10 July 2017

# Importance of the Saharan Heat Low in controlling the North Atlantic free tropospheric humidity budget deduced from IASI $\delta$D observations

Lacour Jean.-Lionel [1,4], Flamant Cyrille [1], Risi Camille [2], Clerbaux Cathy [1,3], and Coheur Pierre-François. [3]

[1]UPMC Univ. Paris 06; Université Versailles St-Quentin, LATMOS-IPSL, Paris, France
[2]UPMC Univ. Paris 06; UMR8539, CNRS/INSU, LMD-IPSL, Paris, France
[3]Spectroscopie de l'Atmosphère, Service de Chimie Quantique et Photophysique, Université Libre de Bruxelles (ULB), Brussel, Belgium
[4]Institute of Earth Sciences, University of Iceland, Reykjavik, Iceland

*Correspondence to:* Jean-Lionel.Lacour@latmos.ipsl.fr

**Abstract.** The isotopic composition of water vapour in the North Atlantic free troposphere is investigated with IASI measurements of the D/H ratio ($\delta$D) above the ocean. We show that in the vicinity of West Africa, the seasonality of $\delta$D is particularly strong (130‰), which is related with the installation of the Saharan Heat Low (SHL) during summertime. The SHL indeed largely influences the dynamic in that region by producing deep turbulent mixing layers, yielding a specific water vapor isotopic footprint. The influence of the SHL on the isotopic budget is analysed at various time and space scales and is shown to be large, highlighting the importance of the SHL dynamics on the moistening and the HDO-enrichment of the free troposphere over the North Atlantic. The potential influence of the SHL is also investigated at the inter-annual scale as we also report important variations of $\delta$D above the Canary Archipelago region. We interpret the variability of the enrichment, using backward trajectory analyses, in terms of the ratio of air-masses coming from the North Atlantic and air-masses coming from the African continent. Finally, the interest of IASI high sampling capabilities is further illustrated by presenting spatial distributions of $\delta$D and humidity above the North Atlantic from which we show that the different sources and dehydration pathways controlling the humidity can be disentangled thanks to the added value of $\delta$D observations. More generally, our results demonstrate the utility of $\delta$D observations obtained from the IASI sounder to gain insight into the hydrological cycle processes in the West African region.

## 1 Introduction

In the North Atlantic, the free tropospheric humidity is the result of a complex interplay between moistening and dehydrating processes of air parcels originating from different sources. While the large scale subsidence largely controls the dryness of the subtropics, numerous other processes have

been shown to moisten the subsiding air, such as large scale transport from the tropics (Sherwood, 1996) or vertical mixing associated with convection, or evaporation of condensate in the convective towers (Sun and Lindzen, 1993). Due to the difficulty to disentangle the relative contributions of these processes and of the different sources, controls on the tropospheric humidity remain imprecise. Additionally, these different sources and processes may be affected by the modulation of regional environmental influences such as the migration of the Inter Tropical Convergence Zone (ITCZ) (Wilcox et al., 2010), the activity of the Saharan Heat Low (SHL) (Lavaysse et al., 2010) or the West African monsoon and associated mesoscale convective systems. Furthermore, the subtropical North Atlantic is a particularly climate sensitive area (Spencer and Braswell, 1997) as the radiative forcing associated with changes in water vapour is the strongest in the free troposphere (Held and Soden, 2000). Efforts in understanding the controls on the North Atlantic humidity are thus crucial.

The measurement of water vapour isotopologues has proved to be a helpful observational diagnostic to study the atmospheric moistening/dehydrating processes in a novel way (e.g. Worden et al. 2007; Frankenberg et al. 2009; Risi et al. 2012; Yoshimura et al. 2014; Tuinenburg et al. 2015; Galewsky et al. 2016). This is because the water isotopologues (HDO, $H_2^{16}O$, $H_2^{18}O$) preferentially condense/evaporate during the phase changes of water, and therefore their isotopic ratio is sensitive to key processes of the hydrological cycle such as airmass mixing (Galewsky et al., 2007), convection (Risi et al., 2008), transport (Galewsky and Samuels-Crow, 2015). The observation of water isotopologues in the vapour provide thereby useful information on the processes that affected the air-masses downwind. While the number of $\delta D$ ($\delta D = 1000 * [(HDO/H_2O)/Rsmow - 1]$, Rsmow being the $HDO/H_2O$ ratio in the standard mean ocean waters) measurements has increased this last decade (e.g. Lacour et al. 2012; Worden et al. 2012; Schneider et al. 2016), it is necessary to understand the factors controlling their variations in order to apprehend their utility for studying hydrological processes. With its demonstrated capabilities to provide $\delta D$ measurements in the free troposphere (Schneider and Hase, 2011; Lacour et al., 2012, 2015), the Infrared Atmospheric Sounding Interferometer (IASI) flying onboard MetOp has since a decade a key role in supplying $\delta D$ observations. IASI has the advantage to make about 1.3 millions measurements a day, which almost ensures one measurement everywhere twice a day. Up to now, IASI $\delta D$ retrievals have been sparsely used for geophysical analyses (Bonne et al., 2015; Tuinenburg et al., 2015).

In this study, we use $\delta D$ and humidity profiles retrieved from IASI at ULB/LATMOS (Lacour et al., 2012, 2015) to explore the isotopic signal at various time and space scales above the North Atlantic near the West African coast and we interpret their seasonal to inter-annual variability as well as their spatial variations. This enables us to investigate the potential of such observations to improve our understanding of the moistening processes in this region. Because the retrieval of $\delta D$ above deserts is difficult due to uncertainties in the surface emissivity and the presence of dust, we have chosen to not analyse the measurements above the Sahara. Nevertheless, because of the

integrated nature of $\delta$D, we show that some information can be derived from $\delta$D signature of air

parcels coming from the desert. Former studies have already been dedicated to the interpretation of isotopic variations observed in precipitation and in water vapour in West Africa (Frankenberg et al., 2009; Risi et al., 2010; Okazaki et al., 2015) by combining models with observations, but solely focusing on the role of convection. From in situ measurements at Izana, González et al. (2016) have shown that different airmass pathways could be detected in $H_2O$-$\delta$D pairs distribution. The

sensitivity of $\delta$D observations to different moisture pathways have also been reported from ground based FTIR and IASI measurements (Schneider et al., 2015) within the MUSICA project (Schneider et al., 2016). Here, from IASI $H_2O$ and $\delta$D ULB/LATMOS retrieval products, we use this property of isotope to first show that the Saharan Heat Low (SHL) - which is a key component of the West African Monsoon system - has a large influence on the budget of water isotopologues above the

North Atlantic in summertime, when the SHL is most active, leading to a strong seasonality of $\delta$D. Then, we present inter-annual variability of the isotopic composition above the Canary Archipelago Region (CAR) and analyse the causes of the variability. Finally, we detail the spatial variations of water vapour and its isotopic composition above the North Atlantic for July 2012 and discuss the different sources and dehydration pathways controlling the free tropospheric humidity.

In section 2 we present the IASI datasets and the different numerical weather prediction model re-analysis that we have used and we provide some guidance on the interpretation of $\delta$D-humidity variations. We analyse the seasonal and inter-annual $\delta$D variations observed above the CAR in sections 3 and 4, respectively. Then in section 5, we describe the spatial distribution of $\delta$D observed in July 2012. Finally, the results are discussed in the conclusion section.

## 2  Data and methods

### 2.1  IASI $\delta$D retrievals

This study is mainly based on $H_2O$ and $\delta$D profiles derived from IASI radiances measurements (Lacour et al., 2012, 2015) from the retrieval processor developed at ULB/LATMOS. IASI is a Fourier Transform spectrometer flying onboard the MetOp platform measuring the thermal infrared emis-

sion of the Earth and the atmosphere (Clerbaux et al., 2009). The high quality spectra (good spectral resolution -0.5cm$^{-1}$- and low radiometric noise) allow to retrieve information on $H_2O$ and $\delta$D in the troposphere after an inversion procedure following the optimal estimation method (Rodgers, 2000) adapted for the requirements of $\delta$D retrieval (Worden et al., 2006; Schneider et al., 2006). With respect to supplying $\delta$D observations in the free troposphere, IASI is the unique successor of the

Tropospheric Emission Spectrometer (TES) instrument which has been used in many isotopic applications studies (i.e. Worden et al. 2007; Risi et al. 2010, 2013). IASI with its high spatio temporal sampling (a measurement almost everywhere on the globe, twice a day) is of great interest for stud-

ies on short terms variations of $\delta$D (Bonne et al., 2015; Tuinenburg et al., 2015) and for an optimal sampling of $\delta$D natural variability (Lacour et al., 2015).

The $\delta$D profiles retrieved from IASI have limited information on the vertical, with degrees of freedom (dofs) varying between 1 and 2 depending on the local conditions (thermal contrast, temperature and humidity profiles, e.g. Pommier et al. 2014). In general, the maximum of sensitivity of the retrieval lies in the free troposphere between 3 and 6 km. For our analysis we use only the retrieved $\delta$D profiles that have more than 1.5 degrees of freedom and that yield a maximum of sensitivity between

4 and 6 km. It is also important to mention that the $\delta$D and humidity retrieved profiles are not exactly representative of the same atmosphere, the humidity profile having more vertical information than $\delta$D. It is thus important to keep in mind that when $\delta$D-$q$ pairs are considered, the $\delta$D estimate is representative of a thicker layer than the $q$ estimate. On an individual basis, the observational error on $\delta$D between 4 and 6 km has been estimated and cross-validated to 38‰ (Lacour et al., 2015). When

several retrieved values are averaged (from N measurements), this error is reduced by a factor $\sqrt{N}$. Because of the large number of IASI measurements, there is presently no near-real-time processing of IASI radiances for $\delta$D. The availability of this quantity is thus limited. In this study we use three different datasets to investigate the isotopic characteristics of water vapour in the North Atlantic:

- a 5 year (2009-2013) dataset above the CAR (26°N-30°N, 18°W-14°W) with an average of
65 measurements available per day;

- a 1 year dataset (2011) along a latitudinal band of narrow longitudinal extent (0°N-60°N, 30°W-25°W), which is used in section 3.4 to evaluate the variations of $\delta$D seasonality along the Western African coast;

- a 1 month dataset (July 2012) above the North Atlantic (0°N-40°N, 40°W-5°W) which is
also used to illustrate the spatial extent of the influence of the SHL (subsection 3.4) and to analyse the different sources and processes controlling the humidity above the North Atlantic in section 5.

These data are used at different time scales from the individual observation to monthly averages. Daily means are obtained by averaging individual observations and monthly averages are obtained

from the daily averages.

There exists another $\delta$D retrieval processor from IASI spectra (Schneider and Hase, 2011) developed within the MUSICA project (Schneider et al., 2016) which we do not use in this study, a brief summary of the differences of the processors is given in the Appendix of Schneider et al. (2016). The MUSICA MetOp/IASI data have already been used previously for documenting the different

moisture pathways in the subtropical North Atlantic region (Schneider et al., 2015).

## 2.2 TES $\delta$D retrieval

In order to derive a climatology of $\delta$D seasonality at a global scale, we also used $\delta$D profiles retrieved from the Tropospheric Emission Spectrometer (TES) measurements (Worden et al., 2012). The TES instrument is, like IASI, a thermal infrared sounder but with a better spectral resolution which makes it more sensitive to the lower troposphere. The observational error on $\delta$D retrieved values from TES has been evaluated to $\pm 30\%o$ (Worden et al., 2012; Herman et al., 2014). The spatiotemporal sampling is however lower than IASI. Nevertheless, $\delta$D retrievals from TES are available since September 2009 and allow for global analysis of $\delta$D.

## 2.3 Backward trajectories analyses and reanalysis data

To help in the interpretation of $\delta$D data we use backward trajectory calculations from the Hybrid Single Particle Lagrangian Integrated Trajectory model (HYSPLIT) (Stein et al., 2015) where NCEP GDAS (Global Data Assimilation System) re-analyses (Kleist et al., 2009) have been used as the meteorological fields. Backward trajectories have been mainly used in the analysis of the dataset above the CAR. Three trajectories arriving at 26°N, 28°N and 30°N at the longitudinal center of the box are computed for each day of IASI observation. In our analysis the trajectories we used are initialized from midday at altitude of 5.5 km. As the retrieved values of IASI are sensitive over a large layer of the true $\delta$D variations, we checked that the air trajectories arriving at 4.5 and 3.5 km were similar. This is shown in Appendix for year 2011. It is also shown that we do not expect temporal mismatch to significantly affect the air parcels history. We also used ECMWF (Dee et al., 2011) and MERRA (Rienecker et al., 2011) re-analysis data to characterize atmospheric dynamics.

## 2.4 Background on $\delta$D analysis

Variations in $\delta$D are to a first order tied to those in absolute humidity (here we use water vapour mixing ratio, noted $q$). For this reason the interpretation of the information contained in $\delta$D is generally done in the $\delta$D-$q$ space, which allows for a joint analysis of their variations. While the interpretation of $\delta$D-$q$ pairs can be complicated as numerous processes can produce a same $\delta$D-$q$ combination, simple models are helpful to understand their position in the $\delta$D-$q$ space (Noone, 2012). The isotopic depletion of water vapour that undergoes condensation at equilibrium can be described by a Rayleigh distillation model as:

$$\delta D = (\alpha - 1)\ln\frac{q}{q_0} + \delta D_0;$$

(1)

with $q_0$ and $\delta D_0$ are the specific humidity and the isotopic composition of the water vapour source, and $\alpha$ is the coefficient of fractionation. This model is shown in Figure 1 for two different sources of water vapour (purple lines). A Rayleigh model with a tropical water vapour source can generally be used to describe the lower limit of the domain of existence of $\delta$D-$q$ pairs. The superior limit of this domain can be described by a mixing model between depleted and dry air from the upper

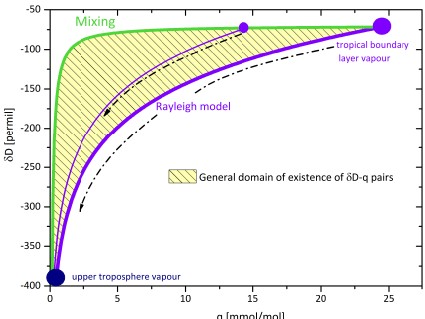

**Figure 1.** Simple models describing the domain of existence of $\delta$D-$q$ pairs. The two purple curves describe the progressive depletion of a tropical boundary layer source and a drier one according to a Rayleigh distillation. The green curve describes the mixing between humidity from the tropical boundary layer source with humidity of the upper troposphere.

troposphere that mixes with enriched and humid air from the tropical boundary layer (green line in Figure 1). The mixing between a source A and B produces a resulting air parcel of mixing ratio q which is the weighted average of the mixing ratio of the two air parcels:

$$q = f[H_2O]_A + (1-f)[H_2O]_B, \tag{2}$$

with $f$, the mixing fraction. The resulting ratio of isotopologues is given as (Galewsky and Hurley, 2010):

$$R_{\mathbf{mix}} = \frac{f[HDO]_A + (1-f)[HDO]_B}{f[H_2O]_A + (1-f)[H_2O]_B}. \tag{3}$$

The mixing model is shown in green in Figure 1. Mixing and distillation of water vapour from different sources can occur over a wide range of combinations and produce $\delta$D-$q$ pairs in between these two boundary models. Noteworthy, intense convective activity act to over deplete water vapour through rain-drop re-evaporation and $\delta$D-$q$ pairs can be found below the Rayleigh distillation model (Worden et al., 2007).

## 3 Seasonal variations: Influence of the SHL on $\delta$D in the subtropical North Atlantic

### 3.1 Seasonal cycle of water vapour and its isotopic composition over the CAR

Figure 2-a shows the composite (2009-2013) seasonal cycles of specific humidity and its isotopic composition at 5.5 km above the CAR. The troposphere is moistened from April to August as the large scale subsidence weakens due to the northward migration of the ITCZ. Then, it progressively dries as the ITCZ retreats south. Interestingly, the $\delta$D composition of water vapour does not follow the exact same cycle as humidity: the air masses are indeed progressively enriched (5‰ per

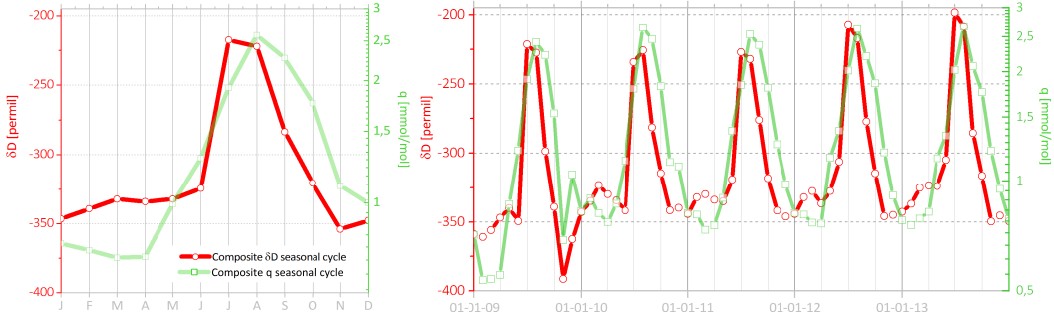

**Figure 2.** a) Composite seasonal cycles of $\delta$D (red) and specific humidity (green) above the CAR for the 2009-2013 time period. b) Monthly variations of $\delta$D (red) and specific humidity (green) above the CAR for the 2009-2013 time period

month) from January to June, before exhibiting an abrupt enrichment from June to July (110‰ in one month). The enrichment persists throughout in August with values nearly as important as in July. Afterwards, the content in HDO rapidly decreases from August to November, by ∼130‰ while humidity stays high until October. This strong seasonality in $\delta$D values is particularly striking and exceeds the seasonality generally found elsewhere (see section 3.4). The period of enrichment in HDO over the CAR appears to coincide with the period when the SHL is present over the Western Sahara as the climatological onset of the SHL occurs at the end of June as the SHL retreats toward the South at the end of September (Lavaysse et al., 2009). In the following, we conduct our analyses with the aim at understanding the factors controlling this strong seasonality and its link with the SHL. Figure 2 (right panel) also shows important inter-annual variability observed in $\delta$D signal not correlated to the humidity variability. We focus on the inter-annual variations in section 4.

## 3.2 The Saharan Heat Low and the associated atmospheric dynamics

In summer, strong heating of the Saharan surface creates a low pressure system (SHL) which enhances convergence in the low levels (see left panel of Figure 3 illustrating the low level circulation for July 2012). In the lower troposphere, the cyclonic circulation around the SHL strengthens simultaneously the south westerly monsoon flow east of its center and the north-easterly Harmattan flow to the west. The near-surface convergence generates enhanced rising motion in an environment prone to dry convection, leading to the formation of deep well mixed boundary layers, whose top often reach 600 hPa or higher. The divergent circulation at the top of the SHL generates an anticyclonic circulation in the middle troposphere, which contributes to the intensification of the African easterly jet (AEJ) further south (Thorncroft and Blackburn, 1999) and which is responsible for the horizontal transport of continental air masses above the Atlantic. The middle tropospheric circulation for July

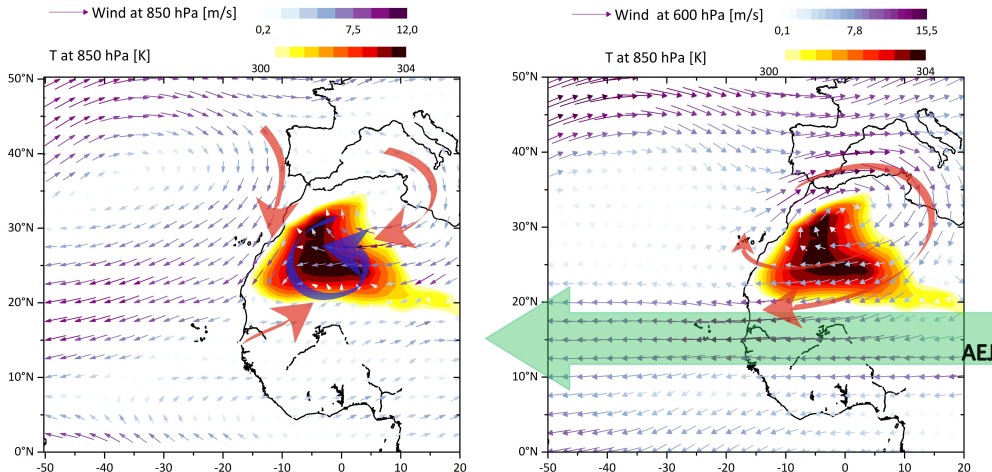

**Figure 3.** Low level (left panel) and mid-level (right panel) circulation associated with the SHL. The location of the SHL is indicated by the temperature field at 850 hPa above 300 K on both panels. The arrows represent wind fields at 850hPa (left panel) and 600hPa (right panel). On the left panel, the red arrows indicate the different sources converging in the depression and the blue arrow represents the anticyclonic circulation associated with the SHL. On the right panel, the red arrows shows how the anti-cyclonic circulation is divided into a part strengthening the AEJ.

2012 represented in Figure 3 shows how the anti-cyclonic circulation contributes to strengthening the AEJ and to bringing mid-level air masses from Northwest Africa over the Northeastern Atlantic.

The development of a heat low due to the surface heating has several consequences on the atmospheric dynamics. In the summer, once the SHL settles in, the top of the Saharan air boundary layer
(SABL) can reach altitude of 6 to 7 km (~550 hPa) due to the enhanced near-surface convergence in the SHL region. The SABL is fully developed in the late afternoon (Chaboureau et al., 2016). During most of the day, while the mixed layer is developing, the characteristics of air masses in the upper part of the SABL (the residual layer) are controlled by advection from the East (Flamant et al., 2007, 2009; Chaboureau et al., 2016). As the warm and dry air moves off the African coast, the SABL
rises and becomes the Saharan Air Layer (SAL) undercut by the cool and moist marine boundary layer. The SAL contributes to important dust transport over the Atlantic (e.g. Prospero and Carlson 1972) and has been widely studied by the aerosol community (e.g. Ben-Ami et al. 2009; Rodríguez et al. 2011).

The intensity of the SHL also has an influence on deep convection over the Sahel (Lavaysse
et al., 2010) and hence on the precipitation over the area. Lavaysse et al. (2010) have indeed shown that during anomalously warm SHL phases (SHL more intense than on average) the southwesterly monsoon flow is reinforced over the Central and Eastern Sahel, leading to enhanced deep convection. At the same time, deep convection is reduced over the Western Sahel. Evan et al. (2015) and Lavaysse

et al. (2016) have further shown that the SHL intensity has increased (due to the warming of the Sahara) in the 2000's with respect to the 1980's, explaining the increase of precipitation observed in Central and Western Sahara in the recent years (Panthou et al., 2014). At inter-annual scales, the intensity of the SHL can significantly modulate deep convection over the Sahel. Furthermore, the diverging anti-cyclonic circulation at the top of the SHL influences the intensity of the AEJ and the westward transport of moisture.

## 3.3 Relationship between the SHL and the summer enrichment over the Atlantic

To gain insights into the processes leading to the seasonal evolution of the isotopic composition of water vapor shown in Figure 2, we analyse the monthly $\delta$D-$q$ diagrams from March to October 2012 in Figure 4-a using Rayleigh and mixing models (the same models than in Figure 1 are shown for simplicity) and using all the IASI observations. The colour indicates the gradient of temperature computed between 5.5 and 1.5 km. In parallel, we provide monthly analyses of airmass trajectories reaching the CAR from HYSPLITT backward trajectory analyses (Figure 4-b). Air masses arriving at the CAR from lower altitudes are identified by dark to light orange lines while air masses coming from higher altitudes are identified by dark to light purple lines. The bottom panels of Figure 4 (Figure 4-c) show the monthly temperature at 850 hPa over the domain, this variable being a proxy of the SHL location over the Sahara.

The $\delta$D-$q$ diagrams in the top panel of Figure 4 show a clear change in the repartition of the $\delta$D-q pairs from May to June. In May, air masses are characterized by low $\delta$D values and a wide range of specific humidity. These observations are localized close to or below the Rayleigh model or close to the dry end of the mixing model. The measurements below the Rayleigh model are associated with air masses coming from the tropics and from lower altitude and can thus be explained by convective processes (e.g. Worden et al. 2007). In June there is a clear separation of the measurements in two clusters, with the cluster of enriched air masses close to the mixing model and the cluster of depleted air masses located between the Rayleigh model and the mixing model. The two clusters also indicate a clear change in the temperature gradients highlighting two different atmosphere. In July, only the enriched cluster is seen with high temperature gradients; it is associated with larger specific humidity values. The same applies in August with slightly more depleted values. In September and October, the observations are located at the left corner of the diagram (more depleted and less humid). The distribution of the $\delta$D-$q$ observations in October 2012 is similar to that of May 2012. This situation then persists until the end of the year (not shown here). In summer, the position of the $\delta$D-$q$ pairs along the mixing model suggest important mixing between dry and depleted air parcels with moist and enriched ones. Additionally, the fact that they are localized on the moist branch of the mixing model indicate a mixing for which the proportion of the moist term is quite important; this is surprising at an altitude of 5.5 km.

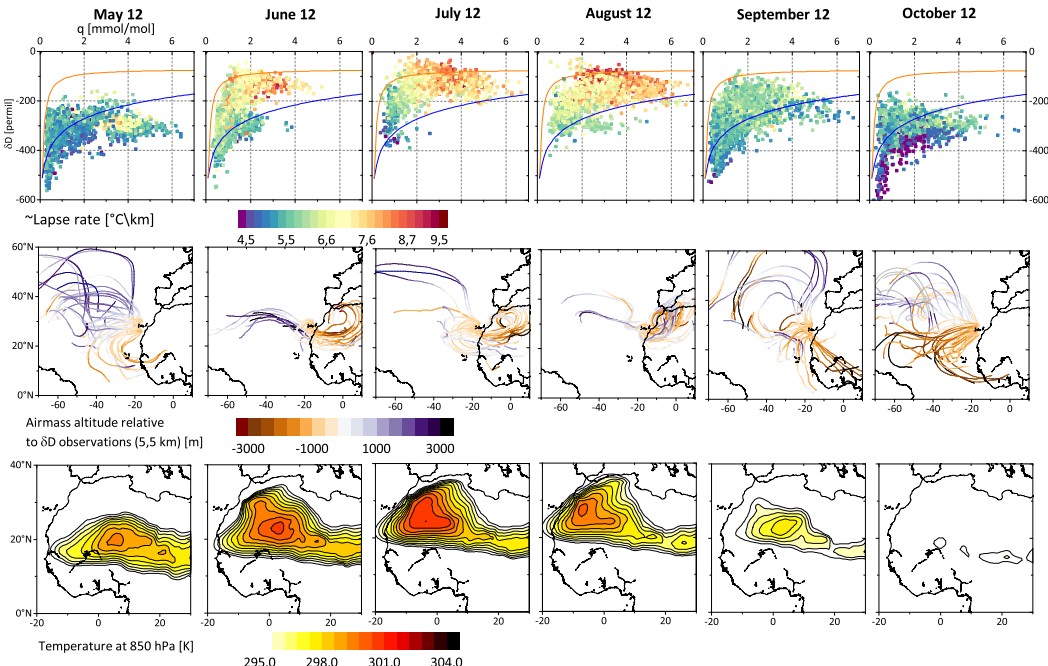

**Figure 4.** Top panels: δD and *q* variations from May to October 2012. All IASI individual observations within the box are shown. The mixing (orange line) and Rayleigh models (blue) are also shown. The temperature lapse rate is given by the colour scale. Middle panels: backward trajectory analyses of the air parcels arriving at the CAR from May to October 2012. The altitude of the air parcels are indicated relatively to the observation altitude (5.5 km). Bottom panels: monthly temperatures at 850 hPa from May to October 2012.

The backward trajectories shown in the middle panel of Figure 4 highlight that the shift in the
cluster of δD-*q* pairs towards higher δD values (from -300 to -100 ‰) corresponds to a change
in the origin of the airmass. During most of the year, the main origin of the air masses arriving
at the CAR is the upper troposphere above the Atlantic Ocean. Conversely, from June to August,
the air masses come from lower altitudes and from a more localized area in the western Sahara. The
situation in June, which is characterized by a HDO-enriched δD-*q* cluster and an HDO-depleted one,
is particularly noteworthy and can be explained with the trajectories analysis: the depleted cluster
would there be associated with air masses coming from the Atlantic and from higher altitudes while
the enriched cluster would correspond to the air masses coming from the African continent and from
lower altitudes. Such differences in the δD-*q* distributions due to different origins of air masses have
already been reported from in situ measurements in González et al. (2016).
The bottom panels of Figure 4 indicate that the change in airmass trajectories is concomitant with
the onset of the SHL, i.e. the installation of the SHL over the Western and central Sahara (Lavaysse
et al., 2009). The concomitant shift of δD-q pairs clusters from a δD value of ∼-300 ‰ to a value
of ∼-100 ‰, with the change of airmass trajectories and the installation of the SHL above the

Western Africa shown here for the year 2012 is observed for the entire 2009-2013 period with some

differences that are discussed in Section 4. Therefore, it appears that the isotopic signal in the water vapour above the Atlantic is strongly influenced by the development of the SHL.

The Figure 5 shows different diagnostics of the state of the atmosphere associated with different combinations of isotopic composition and water vapor content. We plot in Figure 5-a the daily averages (not the individual observations as in Figure 4) $\delta$D-$q$ variations for the period 2009-2013 and

select four boxes defining contrasting conditions: dry and depleted (orange box), dry and enriched (cyan box), humid and depleted (purple box) and humid and enriched (green box). The green box corresponds to the situation found in July-August with air masses coming from the Sahara, the purple box corresponds to depleted values associated with air masses coming from the tropical Atlantic at the end of the summer while the cyan and orange boxes correspond to most of the daily values

found during most of the year and are generally associated with air parcels coming from the North Atlantic. The number of days per month corresponding to each box is plotted in Figure 5-b. The average precipitation amount (computed from the re-analysis at a time step of 3 hours) along the backward trajectories is also shown in Figure 5-c. For each box we plot the composite profiles of the temperature (Figure 5-d), specific humidity (Figure 5-e), relative humidity (Figure 5-f), Richardson

number (Figure 5-g).

For all variables, the ones corresponding to the green and purple boxes strongly deviate from the majority of the observations (characterized by the blue and orange boxes where most of the $\delta$D-$q$ pair lie). The humidity profiles corresponding to the purple box show high humidity (relative and specific) values, which strongly suggest convective processes at play, furthermore confirmed by the

precipitations found along the backward trajectory (Figure 5-c). In the case of the enriched and humid values, corresponding to the summer enrichment (green box), the different profiles characterize a very particular atmosphere. The specific humidity profile has very high values close to the surface, which rapidly decrease with altitude; the relative humidity profile shows also very high values in the first layer of the atmosphere than very dry values up to around 550 hPa; the temperature profile

presents an inversion layer followed by a warm layer; the Richardson number profile indicates very stable values within a thick layer between 900 to 550 hPa. All these features are in fact characteristic of the thermodynamical structure of a deep Saharan air layer (SAL) moving above a thin marine boundary layer. In Figure 6 we show the development of the deep SABL during the year. One can see that the height of the boundary layer reaches the altitude of IASI sensitivity from June to the end

of August. It is however only by the end of June when the air masses originating from the upper part of the ABL above the Sahara are transported above the Atlantic that the $\delta$D signal above the CAR rapidly increases. The seasonality of $\delta$D observed in the free troposphere above the CAR can thus be explained by the development of deep ABL associated with the settling of the SHL over the Western Sahara. The latter acts to efficiently mix dry upper tropospheric air with air from the boundary layer.

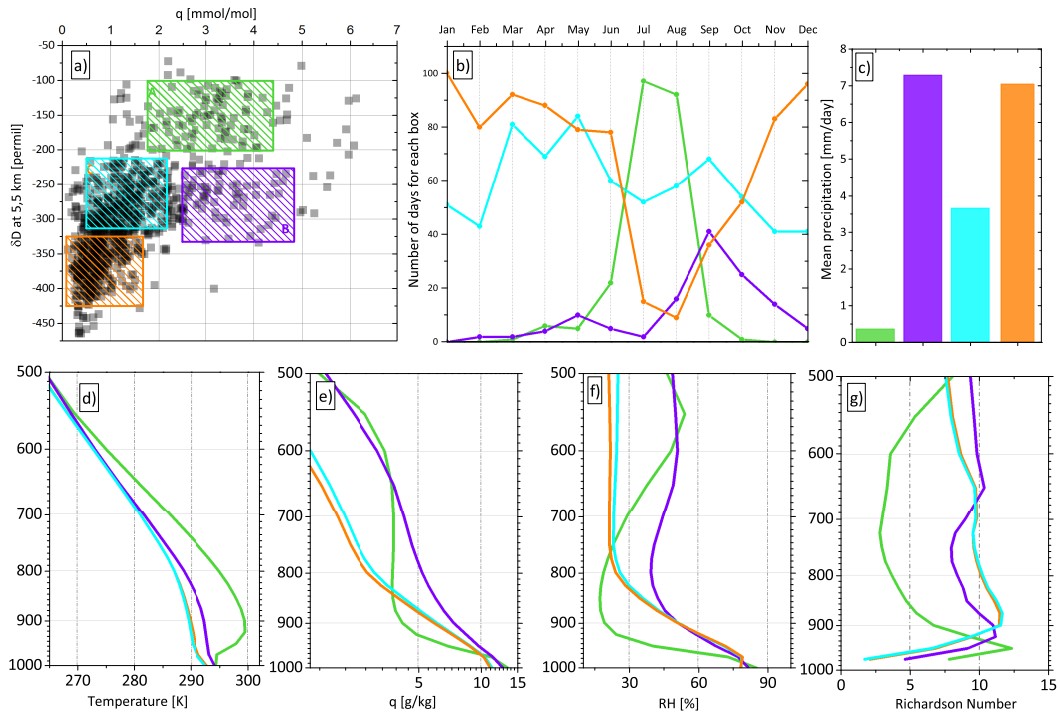

**Figure 5.** Dynamical characterization of the $\delta$D-$q$ space. a) Daily variations of $\delta$D and $q$ for the entire 2009-2013 period with different boxes where composite profiles have been derived from different geophysical parameters (see text for details). b) Number of days per month found within each box, c) mean precipitation along the backward trajectories, panels d) to h) show composite profiles of temperature (K), specific humidity (g/kg), relative humidity, richardson number (from MERRA re-analysis).

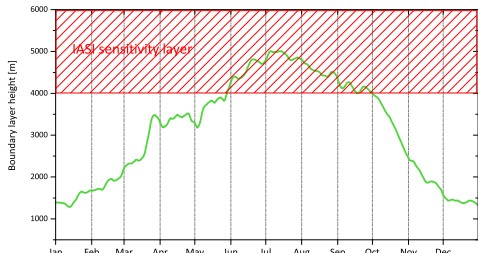

**Figure 6.** Development of deep boundary layers during summer above the Sahara (7°W-5°E,20°N-30°N). Boundary layer heights are extracted from ECMWF ERA re-analysis and are averaged from 2009 to 2013.

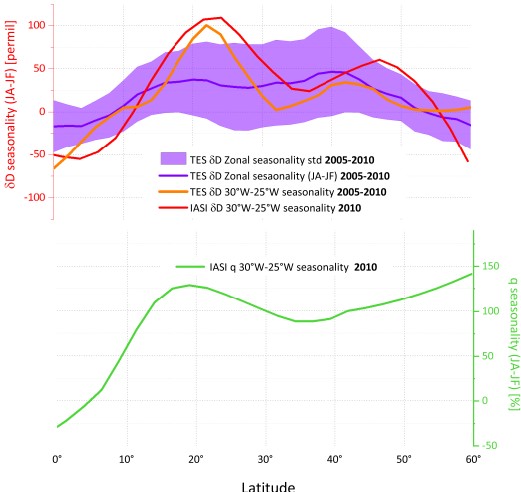

**Figure 7.** Top panel: Global $\delta$D zonal seasonality (JA-JF) from 0 to 60°N (purple line) with its associated standard deviation (shaded area) and seasonality found offshore of West Africa (30°W-25°W,0-60°N) (orange line) observed by TES for the 2005-2010 period and as observed by IASI seasonality for 2010 (red line). Bottom panel: IASI specific humidity seasonality.

Noteworthy, the four boxes delimiting the different $\delta$D-$q$ signatures (here defined arbitrarily) are very similar to the dissociation of in situ $\delta$D-$q$ measurements by González et al. (2016) based on the temperature of the last condensation of air parcels.

### 3.4 Spatial extent of the SHL influence

As the high seasonality in the water isotopic composition observed at the CAR is closely associ-
ated with the activity of the SHL, it can serve as a diagnostic to evaluate the spatial influence of the SHL. In Figure 7-a we first present the seasonality in $\delta$D signal (defined as $\bar{\delta}\mathrm{D}_{\mathrm{July-August}}$ - $\bar{\delta}\mathrm{D}_{\mathrm{January-February}}$) observed from the TES instrument. We use the TES $\delta$D data here as they are available over a longer time period than the IASI dataset considered here and at a global scale. We plot the seasonality of $\delta$D as a zonal mean with its associated standard deviation, calculated for the
2005-2010 period. The seasonality observed off the Western African coast, on an entire latitudinal band of narrow longitudinal extent (30°W-25°W), is also drawn in orange. The latter exhibits a sharp maximum around 22°N, which exceeds values found globally. We attribute this high seasonality as the result of the SHL activity and therefore suggest its influence on the isotopic budget of water vapor extends over a large part of the North Atlantic. Figure 7-b shows the same as Figure 7-a but for IASI
data in 2010. These also show the enrichment in July August from 15° to 30°. The bottom panel of Figure 7, which shows the seasonality for the specific humidity (in percent), reveals a different behaviour. The observed maximum in $\delta$D which does not correspond to the maximum of humidity can also be interpreted as the signature of the SHL, as mixing processes produce a stronger isotopic

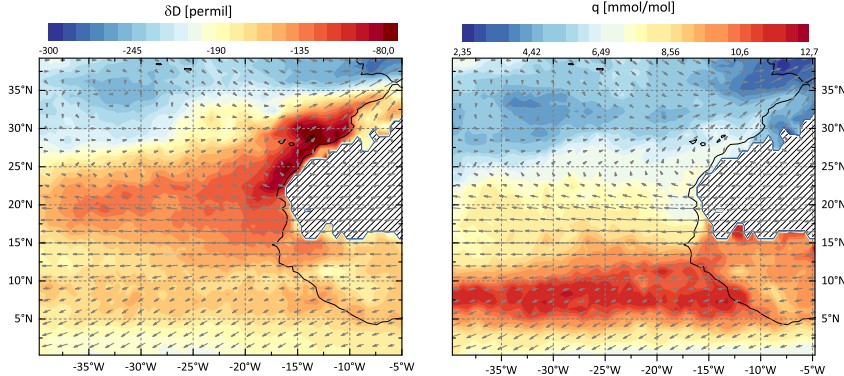

**Figure 8.** $\delta$D (left panel) and $q$ (right panel) distributions from IASI at 4.5 km for July 2012 together with the averaged wind fields at 600 hPa.

signal for a given specific humidity than any other hydrological processes (Galewsky and Hurley, 2010; Noone, 2012).

The differences between humidity and $\delta$D are also clearly visible in the spatial distributions of $\delta$D and $q$ for July 2012 shown in Figure 8. The water vapour distribution strongly differs from its isotopic composition as the maximum in Deuterium enrichment does not appear along the ITCZ, where high $\delta$D values are generally associated with high humidity in convective areas (Risi et al., 2012), where convection act to bring enriched air masses at higher altitudes. Instead, we find the maximum of enrichment further North, around 20°N at the northern edge of the AEJ, for a wide range of specific humidity values.

The spatial pattern drawn by the high seasonality of $\delta$D can be linked to the dynamics of the SAL. Tsamalis et al. (2013) have shown that the SAL displays clear seasonal cycle (both in latitudinal extent and in vertical structure), using 5 years of data from the space-borne Cloud-Aerosol LIdar with Orthogonal Polarization (Winker et al., 2010). The SAL occurs at higher altitudes and farther north during the summer than during winter. Near the African coastline, the SAL is found between 5°-30°N in summer, its northern edge being observed just north of the Canary Islands. The northern edge of the SAL migrates to 15°N during the winter, and is generally observed to be south of the Canary Islands from September to May (see Figure 2 of Tsamalis et al. 2013). During the summer, the SAL is found to be thicker and higher off the coast of Africa, between 1 and 5 km above mean sea level, while it is observed between 1 and 3 km during the winter (see Figure 4 of Tsamalis et al. 2013), i.e. below the altitude of maximum sensitivity of IASI-derived $\delta$D products.

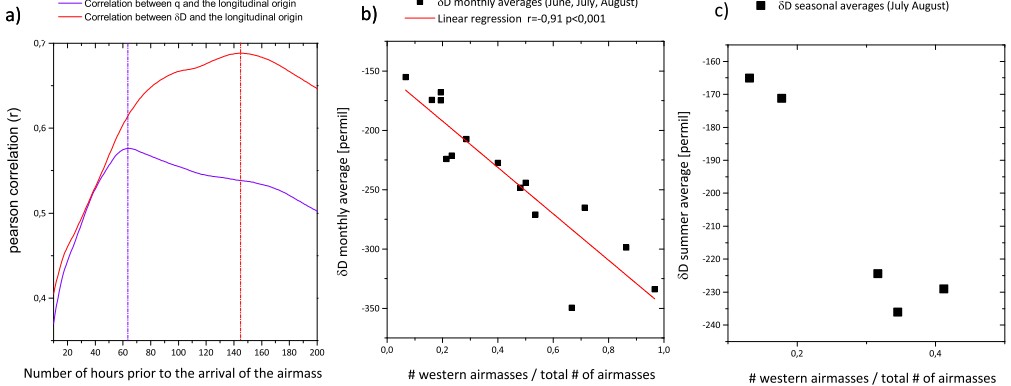

**Figure 9.** a) Correlation between $\delta$D (red curve) and $q$ (purple curve) daily variations and the longitudinal origin of the airmass at various time steps (number of hours prior the arrival of the airmass at the CAR). b) $\delta$D monthly averages (June, July and August) as a function of the ratio of the number of air masses arriving from the Atlantic (Longitude<-20°W) to the total number of air-masses. c) Same as b) but for summer averages (July, August).

## 4    Inter-annual variations above the CAR

Figure 2 shows that there is significant inter-annual variations observed in $\delta$D signal at the CAR. In this section, we investigate the reasons that could explain this variability.

### 4.1    Control of the zonal transport

As explained previously, the $\delta$D variations are sensitive to the source of water vapour. During summertime, the air masses reaching the CAR have contrasting isotopic signatures and water vapour content: the air masses from the Atlantic are dry and depleted, while conversely, the air masses from 350    the African continent are wet and enriched. Schneider et al. (2015) already documented the link between high $\delta$D values and the continental origin of the airmass by coinciding observations of dust concentrations. Thus, the origin of the air masses must control $\delta$D. **?**lso demonstrated the influence of Saharan on $\delta$D by coinciding observations of dust concentrations and $\delta$D-$q$ pairs. In Figure 9-a, 355    we show the correlation between summer (June to August) $\delta$D (and $q$) daily variations and their longitudinal origin for each time step of the backward trajectory analyse. The correlation plot of the $\delta$D (and $q$) daily variations and the longitudinal origin shows that there is a maximum of correlation (r=0.68 with p<0.001), between $\delta$D and the longitudinal origin of the airmass back to 140 hours (6 days) prior to the arrival of the airmass at the CAR. Interestingly, this does not correspond to the 360    same maximum for specific humidity variations, which is 60 hours prior to the arrival. The daily variations of $\delta$D are thus largely controlled by the longitudinal origin of the airmass. The fact that the correlation between $\delta$D and the longitudinal origin is the highest back to 6 days before their ar-

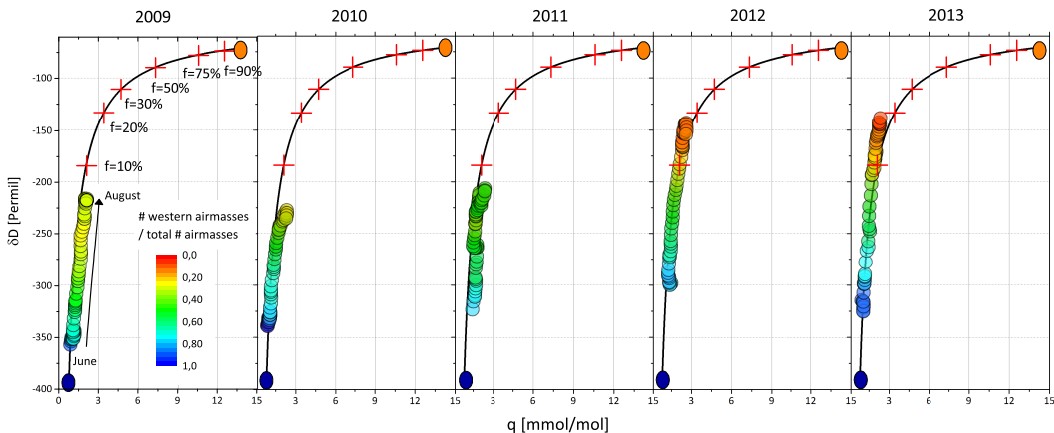

**Figure 10.** Summer enrichments (from June to August) observed at the CAR from 2009 to 2013 along a mixing model defined by the mixing between upper tropospheric water vapor (UT) and boundary layer vapour above the Mediterranean (MBL). IASI daily observations at 5.5 km are smoothed on a 30 days moving average filter and the corresponding ratios of western air-masses to total number of air-masses is shown in colour. The 10, 50, 75 and 90% fractions (f) of MBL of equations 2 and 3 are indicated with the red crosses.

rival compared to 3 days for specific humidity, probably translates the memory effect of $\delta$D. Indeed, while the specific humidity is reset at its saturation value when there is saturation, $\delta$D keeps memory of its history before the saturation (Risi et al., 2010).

In Figure 9-b, we present the monthly variations of $\delta$D in summer as a function of the ratio of western air-masses (air-masses which have a longitudinal origin West of 20°W) to the total number of air-masses arriving at the CAR. A ratio of 1 indicates that the air masses are exclusively coming from the Atlantic while a ratio <<1 indicates that the air-masses mainly come from the continent. We found a linear relation (r=-0.91, p<0.001) between $\delta$D and the ratio indicating that the summer monthly averages of $\delta$D observed at the CAR reflect the balance between the two main origins of the air masses. Logically, this translates into the inter-annual seasonal variations of $\delta$D shown in Figure 9-c. The inter-annual variability of summer $\delta$D values can thus be explained by the relative contributions of air-masses coming from the Atlantic and the ones coming from the SHL.

## 4.2 Control of the mixing fraction

In this section, we assume that the monthly average isotopic composition above the CAR in summer is the result of mixing between upper tropospheric air and boundary layer air, the control of the ratio of the number of the western air-masses to the eastern airmasses can thus be understood in terms of the mixing fraction of the humid source ($f$ in equation 2 and 3) in a mixing model. This is shown in Figure 10 where daily $\delta$D-$q$ pairs from June to August are smoothed on 30 days moving average filter and placed in $\delta$D-$q$ diagrams for each year from 2009 to 2013. The observations show

a progressive moistening and enrichment from June to August along a mixing model. The mixing model used here involves a dry/depleted term with $\delta D=-400‰$ and $q=1$ mmol/mol typical of upper tropospheric air masses that mixes with a moist/enriched term, i.e. water vapour evaporated from the ocean with $\delta D=-70‰$ and $q=14$ mmol/mol (the later being typical for specific humidities above the Mediterranean). Note that a mixing model involving a wetter humid term (with the same isotopic composition) could also fit the observations, this is because $\delta D$ evaporated from the ocean has very similar values and only the specific humidity varies depending on the latitudinal position of the sea. Hence the humid term of the mixing model (in orange in Figure11) could be associated with tropical boundary layer water vapor or boundary layer water vapor from the Mediterranean region (drier) both being potential sources of water vapor fed into the SHL (so called SHL ventilation from south or north, respectively, Lavaysse et al. (2009)). This means that moisture transported at low-level from the oceans and the seas surrounding the continent towards the SHL contribute to the moistening of the free troposphere over the Northeast Atlantic. The SHL is a key player in this process as the relatively moist and enriched air masses are mixed vertically over the depth of the Saharan ABL before being transported over the Ocean due to the divergent, anticyclonic circulation at the top of the SHL. In Figure 10, the colours indicating the ratio of the number of western air-masses to the total number of air-masses show that from June to August, as $\delta D$ increases along the mixing model, the ratio progressively decreases. Assuming constant dry and humid terms, this displacement along the mixing model can be explained by an increase of the mixing fraction (f) in equations 2 and 3. The ratio of the number of western air-masses to the total number of air-masses acts thus like the mixing fraction in controlling the $\delta D$ composition of water vapour. The magnitude of the enrichment is important because in the dry member of the mixing model, a small increase of the fraction of boundary layer air acts to significantly enrich the resulting mixed air-mass (mixing fractions are indicated by the red crosses in Figure 10), conversely, the specific humidity increase is small. The summer enrichment observed at the CAR can therefore be interpreted as the progressive increase of the boundary layer air fraction in the mixing as the SHL acts to efficiently blend boundary layer air over increasing depths from June to August, bringing moisture to altitudes where only upper tropospheric air is observed during the rest of the year. Figure 10 also nicely shows the inter-annual variations observed from 2009 to 2013 as the main origin of the air-masses varies. Assuming constant end members of the mixing, the observations suggest that f reaches up to 20% in 2013 while it is below 10% from 2009 to 2011.

## 5 Dehydration pathways in the North Atlantic

Finally, we have analysed the information contained in the isotopic composition of water vapour over the North Atlantic for July 2012. In Figure 11, we present the $\delta D$ and $q$ distributions previously shown in Figure 8 in the $\delta D$-$q$ space. The IASI $\delta D$-$q$ pairs occupy very distinct domains in the

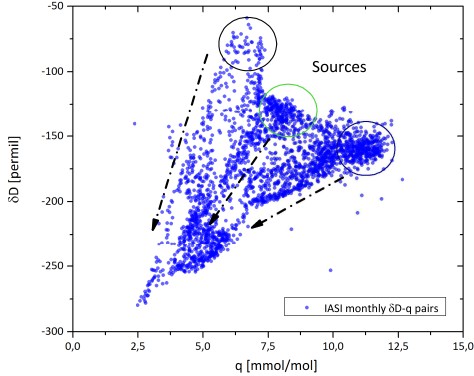

**Figure 11.** $\delta$D-$q$ monthly averages obtained from IASI for July 2012 (see also Figure 8). Each point represents one location over the North Atlantic. Three sources (circles) and different dehydration pathways (black arrows) are identified. See text for details.

diagram indicating a wide variety of sources and processes controlling the water vapour above the Atlantic. In this section we aim at disentangling these various sources and processes. To do this, we map geographically the different sources and pathways identified and use simple models describing
the isotopic depletion of water vapour (Noone, 2012).

### 5.1 Sources

The $\delta$D-$q$ pairs draw distinctive pathways (Figure 11) corresponding to the progressive dehydration/moistening, depletion/enrichment, from different sources. The latter are identified as the moist members of the different branches visually identified of the $\delta$D-$q$ pairs scatter plot. Three different
sources of water vapour have been identified according to their positions in the $\delta$D-$q$ diagrams. These sources are shown in Figure 12-a and mapped on the spatial distribution of $\delta$D in Figure 12-b.

– S1: This source (black squares) is the most enriched and the driest one. It corresponds to water vapour found in a localized area close to the African coast around 26°N and 15°W. We suggest that this enriched source is the direct result of the SHL activity over Sahara. The dry
convective mixing of water vapour from the Mediterranean sea with very dry and depleted air from the upper troposphere - as shown with the mixing model in orange - could indeed produce such high enrichment. Note that the humid term could also be more humid but that this would not affect significantly the mixing model.

– S2: This source (blue diamonds) is the most depleted and the wettest one. It corresponds to
435 water vapour found along the ITCZ between 5° and 10°N. This strongly suggests that this source is the result of the convective uplift of water vapour (with condensation) from lower altitudes. In that case, a Rayleigh model describing the depletion of water vapour evaporated from the tropical ocean can explain the $\delta$D-$q$ values observed (Figure 12-a, purple line).

– S3: This source with intermediate $\delta$D and humidity contents corresponds to a thin longitudinal band along 20°N. This correspond to the Westward transport of dust and aerosol from Africa along the Northern border of the AEJ, where the AEJ is strengthened by the SHL anti-cyclonic circulation (see Figure 3). Comparatively to S1, if we assume that this source is also produced by mixing and that the humid term is similar in both cases, this would mean that the dry term must be different than the mixing potentially explaining S1. As seen in Figure 12-b, the mixing between MBL air that has distilled from tropical water vapor could explain the position of S3 in the $\delta$D-$q$ diagram. We thus hypothesise that S3 is the result of mixing between ascending air from the Sahel (MBL) and air from the AEJ which could be the result of a simple Rayleigh distillation of TBL. Noteworthy, as the S3 exhibit constant $\delta$D and humidity values along the northern border of the AEJ, it means that there is no dehydration and no depletion along this longitudinal band, and thus weak mixing along the westward transport above the Atlantic.

In summary, the air masses that circulated within the divergent flow at the top of the SHL show here two different isotopic signatures depending on whether they contribute to strengthen the AEJ over the continent or if they are transported anticyclonically over the Atlantic around the SHL. These two signatures are distinct from the signature of convection found along the ITCZ.

## 5.2 Pathways

Now that the sources have been identified, we analyse the different moistening and dehydrating pathways visible in the $\delta$D-$q$ space from the different sources (Figure 12-c). To dissociate the different pathways we use their position in $\delta$D-$q$ space and we also use their geographical position to facilitate the dissociation in both spaces. Note that the sources show quasi constant $\delta$D and $q$ values while the dehydration pathways, on the other hand, show an important variability . The dehydration and depletion is mainly latitudinal.

– P1: This pathway describes the dehydration and the rapid depletion of S1. Figure 12-d shows that this pathway corresponds to the area in the North of Africa, dominated by wind (see in Figures 8 and 3) towards Europe where the air is particularly dry and depleted. A mixing model between S1 and a dry term can be used to describe this pathway.

– P2: This pathway corresponds to a small area in between the Canary Islands (S1) and S3. Coherently, in the $\delta$D-$q$ space, these points lie also between S1 and S3. The mixing between S1 and S3 can not explain the observations as a mixing model between these 2 terms would produce a hyperbolic line without reproducing the slope observed. Instead these observations could be explained by the mixing of a constant source such as the MBL with air becoming more enriched and more humid. This could be explained by a stronger influence of the African easterly jet and a weaker influence of the subsidence as we get closer to S3. The Figure 12-e shows how this mechanism could explain the $\delta$D-$q$ pairs observed.

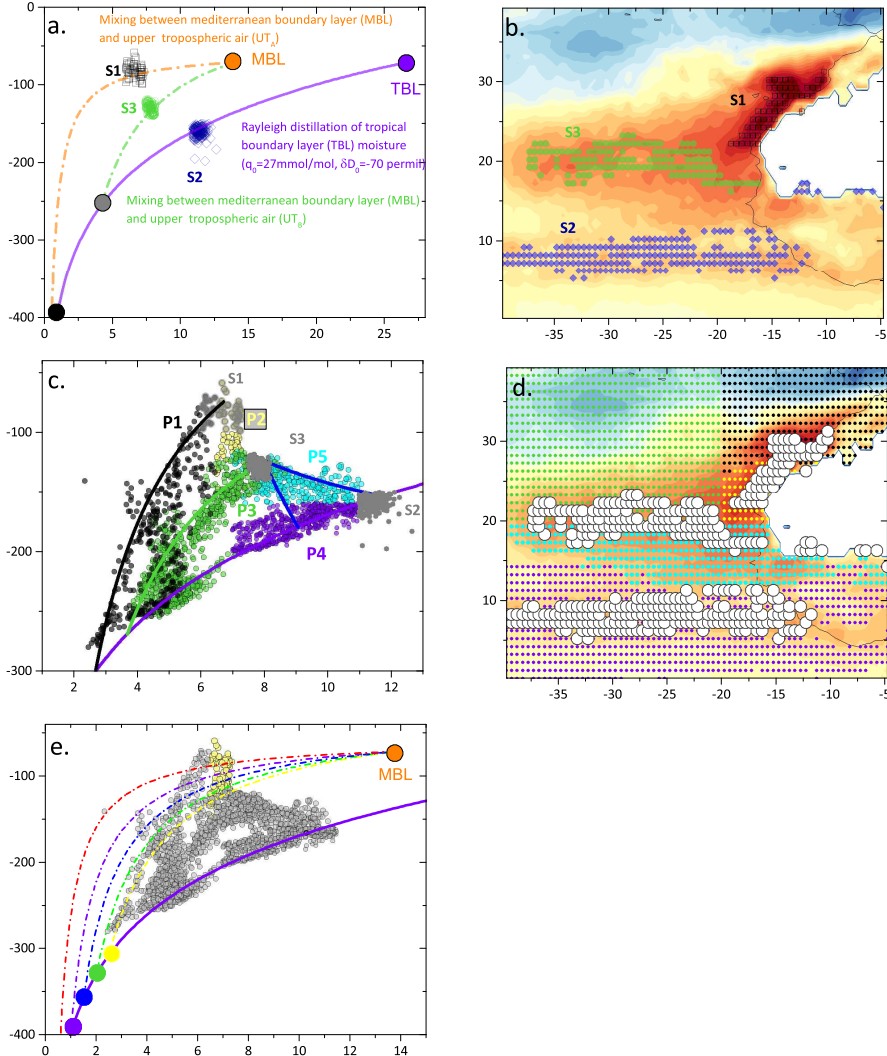

**Figure 12.** a) $\delta$D-$q$ composition of the three main sources identified in the $\delta$D-$q$ diagram with a mixing model (orange curve) between upper tropospheric air (black circle) and Meditteranean boundary layer vapor (orange circle); a Rayleigh model from a tropical boundary layer vapor (purple circle); and a mixing model between MBL and water vapor distilled according to the Rayleigh model defined. b) Geographical location of the different sources identified in a). c) All $\delta$D-$q$ pairs for July 2012 together with different mixing models that could explain the variations observed (see text for details), the colours indicate the different pathways identified. The sources identified in a) are shown in grey. d) Geographical location of the different pathways identified in c). e) Illustration of the mechanism explaining the variations observed in yellow (P2): mixing models between a constant humid term (MBL) and a dry term more and more enriched.

- P3: This pathway corresponds to the western part of the Northern Atlantic where S3 is probably mixed with air from the large scale subsidence as the end member of the mixing model corresponds to the location of the Azores high.

- P4: This pathway describes the depletion of S2 northward and southward of the ITCZ and can be described by a simple Rayleigh distillation model which has the characteristics of the tropical boundary layer moisture.

- P5: This pathway is found between S3 and S2 or corresponds to distilled air parcels from S2. The position of the observations in the $\delta$-$q$ space could be explained again by mixing. However, in that case, the enriched source is drier than the humid source and the corresponding mixing model presents an inverse curvature. The blue curves showing the mixing models computed from S3 and S2 or more distilled terms allows to explain the scatter observed.

The analysis proposed here suggest that the combined observation of water vapor and its isotopic composition can be very useful to identify the different sources of humidity, which are key actors of the hydrological and dynamical cycle of the region, and their interactions. It is however impossible to unambiguously assess the processes responsible of the position of $\delta$D-$q$ pairs as combination of different processes can lead to a same $\delta$D-$q$ position. Nevertheless, the coherence of our interpretation with the actual understanding of the SHL dynamic suggests that the interpretation is reasonable.

## 6  Conclusions

In this study we have explored $\delta$D-$q$ distributions derived from the IASI sounder above the North Atlantic for different time and space scales with the objective of providing an interpretation on the controls of $\delta$D in that region. We have shown that the seasonal enrichment of $\delta$D observed at the CAR was closely linked to the installation of the SHL above the Sahara from June to August. By the end of June, the intense surface heating during the summertime period generates deep boundary layers, which can then be transported above the Atlantic within the so-called SAL. The SAL top reaching the altitude of IASI sensitivity, HDO-enrichment is observed over the CAR. We have shown that the influence of the SHL expands far off the coast, suggesting a large influence of the SHL on the isotopic budget and thus on the humidity budget. The summertime $\delta$-$q$ distributions at the CAR are mainly the result of mixing processes between dry and depleted upper tropospheric air with humid and enriched boundary layer air from the oceans and seas surrounding the West African continent. In the summer, the SHL acts to efficiently mix these contrasting sources and transport anomalously moist and enriched air masses (when compared to the rest of the year) over the Northeast Atlantic Ocean. Inter-annual variations of $\delta$D were also interpreted as the differences in the fraction of western to eastern air-masses arriving at the CAR. The combination of $\delta$D and $q$ observations from IASI in July 2012, together with the knowledge of the key components of the West African Monsoon system,

allowed interpreting the variety of processes driving the water budget over the Northeast Atlantic. More generally this analysis demonstrates the usefulness of $\delta$D measurements from IASI as we show it is possible to disentangle the respective contribution of the different sources of water vapor together with their respective interactions.

The demonstrated capabilities of IASI to provide unique observational constraints on the different sources and processes controlling the free tropospheric humidity in the North Atlantic would be useful to evaluate the representation of these sources and processes in isotopes-enabled climate models. In particular, the strong isotopic signature associated with the SHL and its interactions with the monsoon and the AEJ could be used to assess the correctness of its representation in climate models.

## Appendix A: Supplemental backward trajectory analyses

In this appendix we show additional trajectory analyses we did to verify that there is no spatial and temporal mismatches in the trajectories due to the large sensitivity layer of IASI and due to the differences of time sampling.

### A1 Coherence of airmass trajectories between 3 and 6 kilometers

IASI $\delta$D retrievals are sensitive to $\delta$D variations over a large vertical layer. The information mostly comes from the free troposphere between 3 and 6 km. In Figure A2 we show air-masses arriving at different altitudes (3.5, 4.5 and 5.5 km) within this layer for the year 2011. Airmasses arriving at the different altitudes show similar patterns indicating that what we show at 5.5 km is also valid for the 3-6 km layer.

### A2 Temporal mismatches

Here, we show additional backward trajectory analysis corresponding to different initialization times (9,12 and 21 UTC). The air parcels show very coherent trajectory in between 9 UTC to 21 UTC.

*Author contributions.* J.-L. Lacour did the retrievals of $\delta$D from IASI spectra, performed the data analysis and prepared the manuscript. C. Flamant provided expertise on the SHL and prepared the manuscript with J.-L. Lacour. Camille Risi provided her expertise on the analysis and has corrected the manuscript. P.-F. Coheur has supervised the first part of this study, in relation to the $\delta$D retrievals from IASI. He has corrected the manuscript. C. Clerbaux has supervised the second part of this study and has corrected the manuscript.

*Acknowledgements.* IASI has been developed and built under the responsibility of the "Centre National d'Etudes Spatiales" (CNES, France). It is flown on-board the Metop satellites as part of the EUMETSAT Polar System.

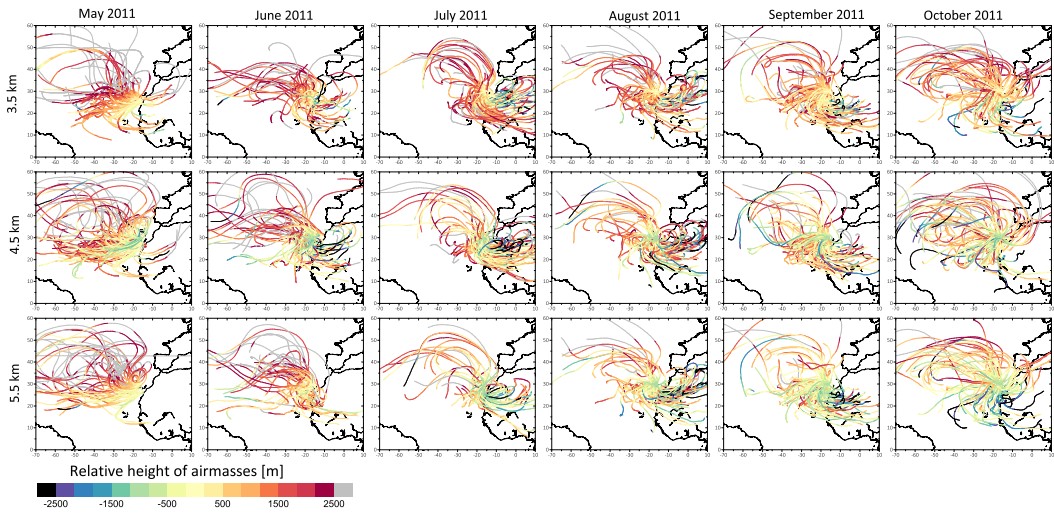

**Figure A1.** Airmasses arriving above the CAR at three different altitudes (3.5, 4.5 and 5.5 km). The trajectories are initialized from 3 points (26°,28° and 30°N and at the longitudinal center of the box at 12.00).

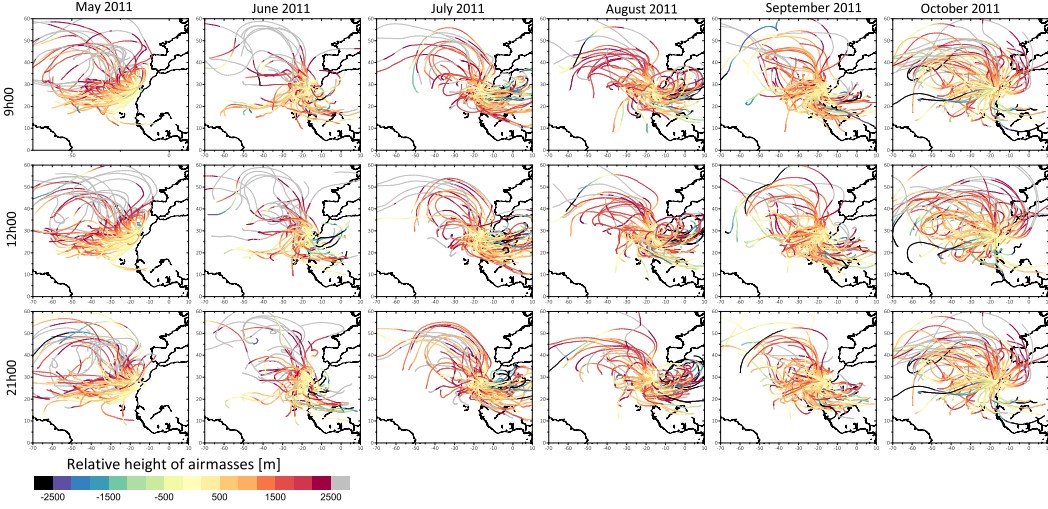

**Figure A2.** Airmasses arriving above the CAR (4.5 km) at three different times (9, 12 and 21 UTC). The trajectories are initialized from 3 points (26°,28° and 30°N and at the longitudinal center of the box).

The IASI L1 data are received through the EUMETCast near real-time data distribution service. Jean-Lionel Lacour is grateful to the CNES for post-doctoral grant. The research in Belgium was funded by the F.R.S.-FNRS, the Belgian State Federal Office for Scientific, Technical and Cultural Affairs (Prodex arrangement 4000111403 IASI.FLOW). Monthly distribution of $\delta$D above the North Atlantic have been retrieved thanks to the support provided by the AC-AHC2 project (ANR-15-CE01-0015). Clerbaux is grateful to CNES for scientific collaboration and financial support. The authors thank the two anonymous referees and Matthias Schneider for their helpful comments to improve the manuscript.

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

-