# Peer review of "Importance of the Saharan Heat Low in controlling the North Atlantic free tropospheric humidity budget deduced from IASI $\delta D$ observations"

_Atmospheric Chemistry and Physics, 2017_

## Referee Comment (RC1) · Anonymous Referee #2 · 30 Mar 2017

In this manuscript, the authors present the isotopic composition of water vapour in the subtropical North Atlantic free troposphere investigated with IASI measurements. This work can be seen as a further step of previous water vapour isotopologues studies carried out in the same region, involving in-situ, ground-based and space-based techniques. In these studies, the observed $H_2O$-δD distribution was characterized as a function of the origin of the airmasses. Here, the authors focus on summer time, where $H_2O$-δD distributions show the mixing between dry and depleted upper tropospheric air with humid and enriched boundary layer air transported within the Saharan air layer. The novelty of the work relies on the identification of the Saharan Heat Low (SHL) as the mechanism controlling the moisture budget in the subtropical Atlantic during summer. This work also shows a simple technique for interpreting the inter-annual variations of δD as a function of the fraction of western to eastern airmasses arriving at Izaña.

Overall, this is a well-written and very interesting manuscript. I recommend publication subject to minor revisions.

**Specific Comments**

The specific comments described below are in relation of a general concern of lack of highlighting previous works developed in this region, which would help to justify the used tools and support the findings.

*SC#1.* Section 2.1 IASI δD retrievals:

The authors use δD IASI retrievals to demonstrate the role of the SHL on the seasonal cycle of the water isotopologue budget above the North Atlantic in summertime. The 5-year ERC Project MUSICA focused on the long-term, global and high-resolution observations of tropospheric $H_2O$-δD. This project used Izaña as a multiplatform site for improving the retrievals of ground-based FTIR and IASI sensors, by comparing with in-situ measurements and airborne profiles. Besides the relevance of the named project on the results of this work, there is no specific mention of it. I recommend including in line 20 of section 2.1, the more recent results of IASI observational errors that can be found in Schneider et al. (2015*, 2016, 2017*), and discuss the use of different approximations for the IASI retrievals.

> \* Schneider et al., Atmos. Meas. Tech., 8, 483–503, doi:10.5194/amt-8-483-2015, 2015
> \*\*Schneider et al., Atmos. Meas. Tech., 10, 507-525, doi:10.5194/amt-10-507-2017, 2017.

*SC#2.* Section 2.2 TES δD retrievals:

Please include a line describing the observational error for TES retrievals (section 2.2).

*SC#3.* Section 3.1 Seasonal cycle of water vapour and its isotopic composition over Izaña:

Is the composite seasonal cycle representative of different years? Are this data in agreement with the inter-annual variability observed with the in-situ records at Izaña? Please, check and discuss.

*SC#4.* Section 3.3 Relationship between the SHL and the summer enrichment over the Atlantic:

The four clusters described in the δD-q distribution plot (Figure 5a) were already observed in the situ-measurements at Izaña. Please, link and discuss the observations.

I would also like to see some references on the discussion on the dynamics of the Saharan Air Layer.

*SC#6.* Please correct typo in page 15, line 17. It is read "2103", instead of "2013.

---

## Referee Comment (RC2) · Anonymous Referee #1 · 12 Apr 2017

In this manuscript, satellite observations of the isotopic composition of free-tropospheric water vapor are used to investigate the processes shaping the moisture budget over the subtropical North Atlantic in summer. The study highlights the importance of the Saharan Heat Low in facilitating the uplift and westward transport of moisture from the continental boundary layer to the oceanic free troposphere. The isotope data are used to shed light on seasonal, interannual and spatial variations in the associated moisture transport and mixing processes. In my view, this is a convincing study that provides important mechanistic insights into the subtropical water cycle and demonstrates the usefulness of isotope observations for such process investigations. I still have quite a few comments that mainly relate to the presentation of the methods

and results, which in my opinion could be improved at several places. Nevertheless, most of these comments should be easy to resolve for the authors. Note that the manuscript contains several minor language errors, and I do not attempt to list all of them (I think these could be eliminated in the copyediting stage).

Specific Comments:

Title: I think the wording of the current title is a bit awkward. What about 'Importance of the Saharan Heat Low in controlling the North Atlantic free tropospheric humidity budget deduced from IASI dD observations'?

Abstract/Introduction: The last part of the abstract (from line 7) and last part of the introduction are a bit unconnected to the rest (read more like a report, first we did .., then . . .). I'd try to improve the connection between the different parts (SHL, interannual and spatial variability).

Abstract: One additional sentence on the more general implications of the work would be good.

Page 2, lines 3-4: 'the dryness of...': I don't understand this. Also in a moist atmosphere the humidity can be variable (even more in absolute terms).

P 2, 21: 'representation' is probably not the correct word; 'understanding'?

P 2, 29-30: 'the seasonal cycle . . . in summertime' is a bit contradictory

P 3, 19: 'filtered based on the residual fit': What does this mean?

Section 2.1: Please add some information on the averaging procedure. For instance, above Izana, do you first calculate daily means by averaging over the individual observations and then monthly means? May there be a bias due to the diurnal cycle? Do you weight the isotope observations by moisture content?

Section 2.3: More details on the trajectory setup would be helpful. At which altitude and time of the day are the trajectories initialized? May there be a bias due to temporal

mismatches between observations and trajectories? Wouldn't it be good to quantify also the uncertainty due to different starting altitudes (by using more than one trajectory per day), since also the satellite observations do represent a vertically extended layer?

P 4, 11: Which reanalysis data set do you use (add reference)?

P 4, 14 and P5, 8: Does q denote specific humidity or mixing ratio? Please use a consistent nomenclature.

P 5, 14-15: 'intense convective activity': I'd be more specific at this point. As I understand the Worden-paper, it is the recycling/evaporation of precipitation that leads to this increased depletion.

P 5, 23: There is also a relatively abrupt increase in q. In my view, the differences in autumn are more pronounced.

Figure 2: From inspecting this figure, the individual values (e.g, for July) shown in panel b do not seem to average to the value shown in panel a. Do you weight by q? Is this really what one should do when calculating such a multi-annual mean value?

P 6, 15: 'before...': I don't understand this insertion. Is it really required? At least you don't need the acronym.

Figure 4: Axis labels should be added to the first row. 'daily variations' is unclear; do you show daily averages or individual observations? Are the Rayleigh and mixing models the same as in Fig. 1 (with the same end members)? Over which levels has the temperature lapse rate been computed? This lapse rate is currently not discussed in the main text (but should be, I think).

P 9, 10: What is the data source for the precipitation amount? Also the reanalysis, which would mean that it's actually a model forecast? What is the accumulation period? This should be mentioned, as it may introduce some uncertainty.

Figure 5: 'Daily variations': see above. Please specify how the Richardson number is

calculated. The vertical velocity is not discussed and could thus be removed. Potential temperature could be shown as an alternative (which would probably also illustrate the deep mixed layer for the green box).

Section 3.4: In my opinion, this section disturbs the flow of the paper. I would shift it after section 4, as it provides a transition to the detailed spatial analysis in section 5.

P 11, 20 – P12, 2: I don't understand this sentence (the connection between the seasonality in q and the mixing processes).

Figure 7: The two upper panels could be combined by adding the red line to the uppermost panel.

Figure 9: The caption says that the ratio in b and c was normalized by the number of air masses from the African continent, but the main text and the axis label suggest that it is normalized by the total number of air masses.

P 14, 7: The wording of the first sentence is unclear.

P 14, 17: 'very similar values' instead of 'always the same value' (it does vary a bit due to changes in SST)

Figure 11: Note that each point represents one location over the North Atlantic. Why do the arrows indicate linear pathways (your simple models describe curved paths in the q-dD space)?

P 16, 12: 'can easily be distinguished': This is a bit subjective. How do you do this? Are the circles just positioned subjectively? Are all data points within the circles shown in panel a of Fig. 12 (this should be explicitly mentioned in the caption)?

P 17, 25: 'processes . . . are horizontal': I don't think that this can be concluded from the present analysis. I'm pretty sure that the descent or ascent of air masses is important for shaping these patterns (as you have demonstrated, e.g., for the SHL).

Figure 12: S2 and S3 are interchanged in panel c. Why are there gaps in the geographical locations of the pathways in the tropics in panel d? More general: Are the pathways defined in geographical or in the q-dD space? Why and how? For instance, in panel c there are some green points (P3) that I would visually attribute to P4.

P 19, 3: Figure 12e instead of 11

Section 5.2: I think some discussion should be added to this section. How unambiguous is the definition of the different pathways? Basically, one could reach every position in the q-dD space (in between your simple models of Fig. 1) by combining different Rayleigh and mixing lines.

---

## Short Comment (SC1) · 13 Apr 2017

The paper presents and interprets δD and {H2O,δD} pair distributions obtained from IASI spectra for the North Atlantic region. It gives interesting insight into the possibilities of such measurements for investigating tropospheric moisture pathways. Similar studies have already been made during the project MUSICA and the respective results are published in several papers: Schneider et al. (2015), Dyroff et al. (2015), González et al. (2016), Schneider et al. (2016). The MUSICA works have been focused on demonstrating the quality and the potential of the remote sensing data whereas this study focuses more on the scientific interpretation of the data. So there are similarities but also clear differences with respect to the MUSICA studies and this new work is very interesting for the scientific community. However, I think it would be important to relate this new work better to the previous MUSICA studies, mention the similarities, and highlight the new aspects.

First, I would very much like to see a statement in the Abstract and/or in the Introduction Section telling the reader that for this new study IASI data generated by the ULB IASI retrieval processor (Lacour et al., 2012; Pommier et al., 2014) are used. In this new study the subtropical North Atlantic moisture pathways are studied for the first time with the ULB IASI retrieval processor data. The here presented data are not generated by the MUSICA MetOp/IASI retrieval processor (Schneider and Hase, 2011; Schneider et al., 2016). The MUSICA MetOp/IASI data have already been used previously for documenting the different moisture pathways in the subtropical North Atlantic region.
The technical details of these retrieval processor differences should maybe not be discussed in an ACP manuscript however, I think it is important to mention that there are different processors. The reason is that the retrieval processor differences can importantly affect the products (Worden et al., 2012, http://www.atmos-meas-tech.net/5/397/2012/): For instance, while the MUSICA processor works with a broad spectral window (Frank and Hase, 2011; Wiegele et al., 2014) similar to the new TES retrieval processor (Worden et al., 2012), the ULB IASI processor fits smaller spectral windows (Lacour et al., 2012). A brief summary of the differences of the processors is given in the Appendix of Schneider et al. (2016).

Second, I would like to recommend setting the here presented data interpretation approaches and the achieved results better in relation to the respective MUSICA activities. In my opinion it would be good to clarify what aspects have already been addressed in the MUSICA papers and what aspects go beyond previous MUSICA works. For example the interpretation of the MUSICA NDACC/FTIR and MUSICA MetOp/IASI {H2O,δD} remote sensing data as shown in Schneider et al. (2015 and 2016) is very similar to what is shown in this new paper in the Sections 3.1, 3.2, and 3.3. Some differences exist in the use of the backward trajectories (in the MUSICA studies the trajectories end at the last condensation point and here they can go beyond the condensation point) and this new work provides an analysis of individual months (taking the summer 2012 as example), whereas the MUSICA studies are mainly limited to an analyses of the overall situation.
Aspects that have not been addressed by the respective MUSICA studies or that have been addressed by using a different approach could then be better highlighted: For instance, Section 3.4 presents a geographically expanded picture if compared to the MUSICA studies and Section 4 shows that quantifying the strength of the Saharan boundary layer mixing signal is possible by a simple backward trajectory analyses whereas in MUSICA the link to the Saharan boundary layer has been documented by coinciding observations of dust concentrations.
The discussion of pathways (Section 5) has similarities to the MUSICA works, but also offers interesting new aspects, like the consideration of a wider geographic region. The clear message from the example study of July 2012 is that such satellite data can be really helpful for investigating geographically varying moisture pathways.

I have furthermore an important remark on the discussion provided in Section 2.4 and on the use of {H2O,δD} pair remote sensing data. As shown in Schneider et al. (2016, and references therein) it is important to ensure that the H2O and δD remote sensing products represent the same vertical altitudes, otherwise defective interpretations of the {H2O,δD} pair distributions are very likely. For this purpose the MUSICA {H2O,δD} pairs are generated by an a posteriori processing (the Type 2 product, which ensures that the {H2O,δD} pair distributions can be correctly interpreted). I think it would be important to clarify how this problem has been addressed for the here presented {H2O,δD} pair data.

I think it would be also important to mention that the transport out of the Saharan boundary layer to the atmosphere above the Canary Archipelago has been studied since many years mainly by the aerosol community (there are leading experts at the Izaña Observatory) and there are a lot of publications available, which I would like to recommend considering (e.g. Rodriguez et al., 2011, http://www.atmos-chem-phys.net/11/6663/2011/ or Rodríguez et al., 2015, http://www.atmos-chem-phys.net/15/7471/2015/, and references therein). Also interesting in this context could be to have a look on the works published for a current ACP/AMT special issue (http://www.atmos-chem-phys.net/special_issue382.html).

As a minor remark I would like to recommend not talking about "data above Izaña" (Izaña is a hill on Tenerife Island and the name of an observatory on this hill). When talking about a region that covers actually the whole Canary Archipelago I would like to recommend using something like "data representative for the Canary Archipelago region".

Finally, I would like to recommend considering for the Introduction Section a reference to the review of Galewski et al. (2016, Rev. Geophys., 54, doi:10.1002/2015RG000512).

---

## Author Comment (AC1) · 9 Jul 2017

In this manuscript, the authors present the isotopic composition of water vapour in the subtropical North Atlantic free troposphere investigated with IASI measurements. This work can be seen as a further step of previous water vapour isotopologues studies carried out in the same region, involving in-situ, ground-based and space-based techniques. In these studies, the observed H2O-$\delta$D distribution was characterized as a function of the origin of the airmasses. Here, the authors focus on summer time, where H2O-$\delta$D distributions show the mixing between dry and depleted upper tropospheric air with humid and enriched boundary layer air transported within the Saharan air layer. The novelty of the work relies on the identification of the Saharan Heat Low (SHL) as the mechanism controlling the moisture budget in the subtropical Atlantic during summer. This work also shows a simple technique for interpreting the interannual variations of $\delta$D as a function of the fraction of western to eastern airmasses arriving at Izaña. Overall, this is a well-written and very interesting manuscript. I recommend publication subject to minor revisions.

The Authors would like to thank the reviewer #2 for reviewing this manuscript and for providing comments. Our answers to referee's comments are shown in blue and changes in text are shown in grey.

 The specific comments described below are in relation of a general concern of lack of highlighting previous works developed in this region, which would help to justify the used tools and support the findings.

SC#1. Section 2.1 IASI $\delta$D retrievals: The authors use $\delta$D IASI retrievals to demonstrate the role of the SHL on the seasonal cycle of the water isotopologue budget above the North Atlantic in summertime. The 5-year ERC Project MUSICA focused on the long-term, global and high-resolution observations of tropospheric H2O-$\delta$D. This project used Izaña as a multiplatform site for improving the retrievals of ground-based FTIR and IASI sensors, by comparing with in-situ measurements and airborne profiles. Besides the relevance of the named project on the results of this work, there is no specific mention of it. I recommend including in line 20 of section 2.1, the more recent results of IASI observational errors that can be found in Schneider et al. (2015*, 2016, 2017*), and discuss the use of different approximations for the IASI retrievals.

* Schneider et al., Atmos. Meas. Tech., 8, 483–503, doi:10.5194/amt-8-483-2015, 2015

**Schneider et al., Atmos. Meas. Tech., 10, 507-525, doi:10.5194/amt-10-507-2017, 2017.

We followed your recommendation concerning the lack of references to the work done within the MUSICA project in that region. More references on previous work related to the sensitivity of $\delta$D to airmasses history are added in the introduction. See also our more detailed reply to Matthias Schneider who also pointed out a lack of  discussion on the MUSICA work.

We also now make sure that it is clear for the reader that we use the retrieval of  $\delta$D  from IASI developed at ULB/LATMOS and not the MUSICA one. Concerning the description of IASI error, we thus only refer to the corresponding error characterization works (Lacour et al.,2012 and Lacour et al., 2015).

SC#2.

Section 2.2 TES $\delta$D retrievals: Please include a line describing the observational error for TES retrievals (section 2.2).

Added:

The observational error on δD retrieved values from TES has been evaluated to 30‰ (Worden et al., 2012; Herman et al., 2014).

SC#3. Section 3.1 Seasonal cycle of water vapour and its isotopic composition over Izaña: Is the composite seasonal cycle representative of different years? Are this data in agreement with the inter-annual variability observed with the in-situ records at Izaña? Please, check and discuss.

Yes the composite is realized from 5 years of IASI data (2009-2013). However, it is difficult to compare with in situ data at Izana since the ground-based data described in Gonzalez et al.( 2015) is for the years 2012 to 2013 at one site and 2013 to 2015 for another site, hence with little temporal overlap with our dataset.. Moreover Gonzalez et al. (2015) showed there was an important diurnal cycle of δd due to the development/displacement of the boundary layer with night measurements being the most representative of the free troposphere. A comparison of in-situ measurements with IASI ones is thus not straightforward and would require the development of a cautious frame to do so properly. Nevertheless, this would be a work of interest.

SC#4. Section 3.3 Relationship between the SHL and the summer enrichment over the Atlantic: The four clusters described in the δD-q distribution plot (Figure 5a) were already observed in the situ-measurements at Izaña. Please, link and discuss the observations. I would also like to see some references on the discussion on the dynamics of the Saharan Air Layer.

Referee is right, this is a link we should have made.

Noteworthy, the four boxes delimiting the different δD-q signatures (here defined arbitrarily) are very similar to the dissociation of in situ δD measurements by Gonzales et al., 2016 based on the temperature of the last condensation of air parcels.

We also add a discussion on the SAL:

The spatial pattern drawn by the high seasonality of δD can be linked to the dynamics of the SAL. Tsamalis et al. (2013) have shown that the SAL displays clear seasonal cycle (both in latitudinal extent and in vertical structure), using 5 years of data from the space-borne Cloud-Aerosol Lidar with Orthogonal Polarization (Winker et al., 2010). The SAL occurs at higher altitudes and farther north during the summer than during winter. Near the African coastline, the SAL is found between 5-30°N in summer, its northern edge being observed just north of the Canary Islands. The northern edge of the SAL migrates to 15_N during the winter, and is generally observed to be south of the Canary Islands from September to May (see Figure 2 of Tsamalis et al. 2013). During the summer, the SAL is found to be thicker and higher off the coast of Africa, between 1 and 5 km above mean sea level, while it is observed between 1 and 3 km during the winter (see Figure 4 of Tsamalis et al. 2013), i.e. below the altitude of maximum sensitivity of IASI-derived δD products.

SC#6. Please correct typo in page 15, line 17. It is read "2103", instead of "20

Corrected, thank you.

---

## Author Comment (AC2) · 9 Jul 2017

**In this manuscript, satellite observations of the isotopic composition of free tropospheric water vapor are used to investigate the processes shaping the moisture budget over the subtropical North Atlantic in summer. The study highlights the importance of the Saharan Heat Low in facilitating the uplift and westward transport of moisture from the continental boundary layer to the oceanic free troposphere. The isotope data are used to shed light on seasonal, interannual and spatial variations in the associated moisture transport and mixing processes. In my view, this is a convincing study that provides important mechanistic insights into the subtropical water cycle and demonstrates the usefulness of isotope observations for such process investigations. I still have quite a few comments that mainly relate to the presentation of the methods and results, which in my opinion could be improved at several places. Nevertheless, most of these comments should be easy to resolve for the authors. Note that the manuscript contains several minor language errors, and I do not attempt to list all of them (I think these could be eliminated in the copyediting stage).**

We are grateful to Referee #1 for his/her positive and attentive review. His/her numerous comments have been very useful to improve the manuscript. Our answers to referee's comments are shown in blue and changes in text are shown in grey.

Specific Comments:

Title: I think the wording of the current title is a bit awkward. What about 'Importance of the Saharan Heat Low in controlling the North Atlantic free tropospheric humidity budget deduced from IASI dD observations'?

We now follow your suggestion, thank you.

Abstract/Introduction: The last part of the abstract (from line 7) and last part of the introduction are a bit unconnected to the rest (read more like a report, first we did .., then . . .). I'd try to improve the connection between the different parts (SHL, interannual and spatial variability).

Changes have been made to improve the connection.

Abstract: One a²dditional sentence on the more general implications of the work would be good.

Added:

> "More generally, our results demonstrate the utility of $\delta$D observations obtained from the IASI sounder to gain insight into the hydrological cycle processes in the West African region."

Page 2, lines 3-4: 'the dryness of...': I don't understand this. Also in a moist atmosphere the humidity can be variable (even more in absolute terms).

We meant to refer to the logarithmic dependence of the OLR to changes in specific humidity where in dry area a small change of specific humidity has a great influence on the OLR (greater than a small change in humid areas). But this is now removed for sake of simplicity.

>

P 2, 21: 'representation' is probably not the correct word; 'understanding'?

We changed to 'understanding'

P 2, 29-30: 'the seasonal cycle . . . in summertime' is a bit contradictory

This has been changed to:

> "(..) the Saharan Heat Low (SHL) - which is a key component of the West African Monsoon system - has a large influence on the budget of water isotopologues above the North Atlantic in summertime, when the SHL is most active, leading to a strong seasonality of δD."

P 3, 19: 'filtered based on the residual fit': What does this mean?

This is related to the retrieval procedure that requires the fitting of computed spectra on the measured spectra. The residual is the difference between the final computed spectra and the measured one. The sentence needs more details to be properly understood and is not of prime importance, we thus removed it

>

Section 2.1: Please add some information on the averaging procedure. For instance, above Izana, do you first calculate daily means by averaging over the individual observations and then monthly means? May there be a bias due to the diurnal cycle? Do you weight the isotope observations by moisture content?

We now add information on the averaging. IASI overpasses are around 9.30 AM and 9.30 PM local time and are likely to induce a bias if there is a strong diurnal cycle.

> "These data are used at different time scales from the individual observation to monthly averages. Daily means are obtained by averaging individual observations from morning and evening IASI measurements which is likely to introduce a bias if there is a diurnal cycle. Monthly averages are obtained from the daily averages."

Section 2.3: More details on the trajectory setup would be helpful. At which altitude and time of the day are the trajectories initialized? May there be a bias due to temporal mismatches between observations and trajectories? Wouldn't it be good to quantify also the uncertainty due to different starting altitudes (by using more than one trajectory per day), since also the satellite observations do represent a vertically extended layer?

More details are now given. Referee is right mentioning potential temporal and spatial mismatches between observations and trajectories. We thus tested if trajectories arriving at different altitudes representative of the IASI sensitivity layer have similar patterns and we also tested the temporal differences. The outcome of this test is that the situation presented is generally valid. We provide the different trajectory analyzes in appendix.

P 4, 11: Which reanalysis data set do you use (add reference)?

This is now specified:

> (..) we use backward trajectory calculations from the Hybrid Single Particle Lagrangian Integrated Trajectory model (HYSPLIT) (Stein et al., 2015) where NCEP  GDAS (Global Data Assimilation System) re-analyses (Kleist et al., 2009) have been used as the meteorological fields

P 4, 14 and P5, 8: Does q denote specific humidity or mixing ratio? Please use a consistent nomenclature.

q is the mixing ratio. This has been corrected.

P 5, 14-15: 'intense convective activity': I'd be more specific at this point. As I understand the Worden-paper, it is the recycling/evaporation of precipitation that leads to this increased depletion.

This has been changed to:

> Noteworthy, intense convective activity act to over deplete water vapor through rain-drop re-evaporation and δD-q pairs can be found below the Rayleigh distillation model (Worden et al., 2007).

P 5, 23: There is also a relatively abrupt increase in q. In my view, the differences in autumn are more pronounced.

The now corrected Figure 2 (see next answer) better highlights a difference between the enrichment (in June) and the progressive moistening (from April).

Figure 2: From inspecting this figure, the individual values (e.g, for July) shown in panel b do not seem to average to the value shown in panel a. Do you weight by q? Is this really what one should do when calculating such a multi-annual mean value?

Thank you for the in-depth inspection of the Figure, there was indeed something wrong. We realized that the monthly averages were not properly done from daily averages. This is now corrected and has been double checked: monthly averages are computed from daily means and composites are derived from monthly averages. The seasonality is thus less pronounced that previously stated, so that this is now also corrected in the revised MS.

P 6, 15: 'before...': I don't understand this insertion. Is it really required? At least you don't need the acronym.

Agreed, we have simplified the discussion here.

Figure 4: Axis labels should be added to the first row. 'daily variations' is unclear; do you show daily averages or individual observations?

We show individual observations, this is now stipulated in the legend and axis labels have been added.

Are the Rayleigh and mixing models the same as in Fig. 1 (with the same end members)? Over which levels has the temperature lapse rate been computed? This lapse rate is currently not discussed in the main text (but should be, I think).

They weren't exactly the same, now they are.

We now add the information on the temperature gradient:

> The colour indicates the gradient of temperature computed between 5.5 and 1.5 km.

And is briefly discussed in the text. But this is more discussed in the following section.

P 9, 10: What is the data source for the precipitation amount? Also the reanalysis, which would mean that it's actually a model forecast? What is the accumulation period? This should be mentioned, as it may introduce some uncertainty.

Yes, the precipitation also come from the reanalysis.

The average precipitation amount (computed from the re-analysis at a time step of 3 hours)

Figure 5: 'Daily variations': see above. Please specify how the Richardson number is calculated. The vertical velocity is not discussed and could thus be removed. Potential temperature could be shown as an alternative (which would probably also illustrate the deep mixed layer for the green box).

It is now stated that these are daily variations and not individual observations. Richardson number is from MERRA re-analysis. We removed the vertical velocity panel and choose to not add the potential temperature for sake of simplicity.

Section 3.4: In my opinion, this section disturbs the flow of the paper. I would shift it after section 4, as it provides a transition to the detailed spatial analysis in section 5.

We agree that it could fit before the spatial analysis (it was there in a previous section) but we prefer to have it in Section 3 "Seasonal variations: Influence of the SHL on _D in the subtropical North Atlantic" since we use the seasonality to derive the spatial influence.

P 11, 20 − P12, 2: I don't understand this sentence (the connection between the seasonality in q and the mixing processes).

The bottom panel of Figure 6, which shows the seasonality for the specific humidity (in percent), reveals a different behaviour. The observed maximum in $\delta$D wich does not correspond to the maximum of humidity can also be interpreted as the signature of the SHL, as mixing processes produce a stronger isotopic signal for a given specific humidity than any other hydrological processes (Galewsky2010,Noone2012).

Figure 7: The two upper panels could be combined by adding the red line to the uppermost panel.

We now combine the two panels.

Figure 9: The caption says that the ratio in b and c was normalized by the number of air masses from the African continent, but the main text and the axis label suggest that it is normalized by the total number of air masses.

Thank you, it is the ratio of #of western air-massess/# total number

P 14, 7: The wording of the first sentence is unclear.

Changed to:

In this section we translate the control of the airmass origins in terms of mixing fraction of the mixing.

P 14, 17: 'very similar values' instead of 'always the same value' (it does vary a bit due to changes in SST)

Yes referee is right, this has been corrected.

Figure 11: Note that each point represents one location over the North Atlantic. Why do the arrows indicate linear pathways (your simple models describe curved paths in the q-dD space)?

The idea of the figure was to show that the δD-q pairs distribution can be decomposed into different pathways from three different points (sources). The arrows have no physical meaning here.

P 16, 12: 'can easily be distinguished': This is a bit subjective. How do you do this? Are the circles just positioned subjectively?

We try to be less subjective:

> "The latter are identified as the moist members of the different branches visually identified of the δD-q pairs scatter plot."

Are all data points within the circles shown in panel a of Fig. 12 (this should be explicitly mentioned in the caption)?

The circles were "drawn by hand" and some contours on q and δD were arbitrarily defined. The contours are now better defined. See also our answer to your last comment.

P 17, 25: 'processes . . . are horizontal': I don't think that this can be concluded from the present analysis. I'm pretty sure that the descent or ascent of air masses is important for shaping these patterns (as you have demonstrated, e.g., for the SHL).

Agreed. This is conclusion was a bit simplistic as the stronger subsidence is likely to contribute to the depletion and drying Northward. This has been removed and we now just mentioned:

> "Note that the sources show quasi constant δD and q values while the dehydration pathways, on the other hand, show an important variability . The dehydration and depletion is mainly latitudinal."

Figure 12: S2 and S3 are interchanged in panel c. Why are there gaps in the geographical locations of the pathways in the tropics in panel d? More general: Are the pathways defined in geographical or in the q-dD space? Why and how? For instance, in panel c there are some green points (P3) that I would visually attribute to P4.

The S2-S3 swap is now corrected, thank you.

The best would have been to dissociate the different pathways from their position in the q-δd diagrams. However to simplify the procedure, we sometimes used the geographical locations of the q-δd pairs. This is why the geographical limits between the different pathways are sometimes sharp. This is now explicitly stated. We could indeed think that some green point are related to P4 however their geographical position prevent us to link them to P4.

P 19, 3: Figure 12e instead of 11

ok

Section 5.2: I think some discussion should be added to this section. How unambiguous is the definition of the different pathways? Basically, one could reach every position in the q-dD space (in between your simple models of Fig. 1) by combining different Rayleigh and mixing lines.

The analysis proposed here suggest that the combined observation of water vapor and its isotopic composition can be very useful to identify the different sources of humidity, which are key actors of

the hydrological and dynamical cycle of the region, and their interactions. It is however impossible to unambiguously assess the processes responsible of the position of $\delta$D-q pairs as combination of different processes can lead to a same $\delta$D-q position. Nevertheless, the coherence of our interpretation with the actual understanding of the SHL dynamic suggest that it is a reasonable interpretation.

---

## Author Comment (AC3) · 9 Jul 2017

**Matthias Schneider short comment**

**The paper presents and interprets δD and {H2O,δD} pair distributions obtained from IASI spectra for the North Atlantic region. It gives interesting insight into the possibilities of such measurements for investigating tropospheric moisture pathways. Similar studies have already been made during the project MUSICA and the respective results are published in several papers: Schneider et al. (2015), Dyroff et al. (2015), González et al. (2016), Schneider et al. (2016). The MUSICA works have been focused on demonstrating the quality and the potential of the remote sensing data whereas this study focuses more on the scientific interpretation of the data. So there are similarities but also clear differences with respect to the MUSICA studies and this new work is very interesting for the scientific community. However, I think it would be important to relate this new work better to the previous MUSICA studies, mention the similarities, and highlight the new aspects.**

We thank Dr. Schneider for having taken the time to comment on our manuscript. We agree with him that similar studies have been conducted in the framework of the MUSICA project. The different seasonal δd-q signatures above Izana have been observed from our IASI retrieval for a few years (i.e. Lacour J.-L. ULB PhD thesis in 2015) and much work has been done trying to understand the link with the dynamics of the region. It is the purpose of the present study to improve our knowledge of the influence of the regional atmospheric dynamics on the water vapor budget over the Northeastern Atlantic, and of the processes leading to the seasonal q-δd distribution. We agree with him that some references on MUSICA work related to the sensitivity of q-δd pairs to moisture pathways in that region were missing and we believe this has been now corrected.

First, I would very much like to see a statement in the Abstract and/or in the Introduction Section telling the reader that for this new study IASI data generated by the ULB IASI retrieval processor (Lacour et al., 2012; Pommier et al., 2014) are used. In this new study the subtropical North Atlantic moisture pathways are studied for the first time with the ULB IASI retrieval processor data. The here presented data are not generated by the MUSICA MetOp/IASI retrieval processor (Schneider and Hase, 2011; Schneider et al., 2016). The MUSICA MetOp/IASI data have already been used previously for documenting the different moisture pathways in the subtropical North Atlantic region. The technical details of these retrieval processor differences should maybe not be discussed in an ACP manuscript however, I think it is important to mention that there are different processors. The reason is that the retrieval processor differences can importantly affect the products (Worden et al., 2012, http://www.atmos-meas-tech.net/5/397/2012/): For instance, while the MUSICA processor works with a broad spectral window (Frank and Hase, 2011; Wiegele et al., 2014) similar to the new TES retrieval processor (Worden et al., 2012), the ULB IASI processor fits smaller spectral windows (Lacour et al., 2012). A brief summary of the differences of the processors is given in the Appendix of Schneider et al. (2016).

We agree that the information on the retrieval processor might not be clear for the reader. This is now clarified in the introduction:

> In this study, we use δD and humidity profiles retrieved from IASI at ULB/LATMOS (Lacour et al., 2012; Lacour et al., 2015).

This is also clearly stated in the data section.

We added the missing references about the identification of moisture pathways from IASI and FTIR MUSICA products.

> From in situ measurements at Izana, González et al. (2016) have shown that different airmass pathways could be detected in H2O-δD pairs distribution. The sensitivity of δD observations to different moisture pathways have also been reported from ground based FTIR and IASI measurements (Schneider et al., 2015) within the MUSICA project (Schneider et al., 2016). Here, we use IASI H2O and δD ULB/LATMOS retrieval products (..)

Second, I would like to recommend setting the here presented data interpretation approaches and the achieved results better in relation to the respective MUSICA activities. In my opinion it would be good to clarify what aspects have already been addressed in the MUSICA papers and what aspects go beyond previous MUSICA works. For example the interpretation of the MUSICA NDACC/FTIR and MUSICA MetOp/IASI {H2O,δD} remote sensing data as shown in Schneider et al. (2015 and 2016) is very similar to what is shown in this new paper in the Sections 3.1, 3.2, and 3.3. Some differences exist in the use of the backward trajectories (in the MUSICA studies the trajectories end at the last condensation point and here they can go beyond the condensation point) and this new work provides an analysis of individual months (taking the summer 2012 as example), whereas the MUSICA studies are mainly limited to an analyses of the overall situation.

> Aspects that have not been addressed by the respective MUSICA studies or that have been addressed by using a different approach could then be better highlighted: For instance, Section 3.4 presents a geographically expanded picture if compared to the MUSICA studies and Section 4 shows that quantifying the strength of the Saharan boundary layer mixing signal is possible by a simple backward trajectory analyses whereas in MUSICA the link to the Saharan boundary layer has been documented by coinciding observations of dust concentrations. The discussion of pathways (Section 5) has similarities to the MUSICA works, but also offers interesting new aspects, like the consideration of a wider geographic region. The clear message from the example study of July 2012 is that such satellite data can be really helpful for investigating geographically varying moisture pathways.

There are indeed similarities with previous work done within MUSICA but we believe the general approach proposed here is somehow different than the previous studies you report and can thus not be compared easily. We went through our manuscript again and made sure the previous works are properly addressed.  Similarities with the work by Gonzalez is also now better addressed.  (See referee #2 comments). Schneider et al. (2015 and 2016) clearly showed the sensitivity of δD to different air parcels trajectories, which is now clearly acknowledged in our manuscript but we do not think it is a good idea to compare every similarity in that study.  In Section 3.1 3.2 and 3.3 of our paper we use the sensitivity of δD to different moisture sources to show that we have a clear signal associated to the arrival of the SHL.  The message of the section 3 is the concomitant signal in δD and the SHL. The use of backward trajectory analyses is frequent in isotope analysis and we believe it is not worth to mention that some differences occur from one backward trajectory analyze to another. We agree with referee that we could link the section 4 with  Schneider et al. (2015) concerning the coincidence of dust and high δd values. So we added:

> Schneider et al., (2015) already documented the link between high δD values and the continental origin of the airmass by  coinciding observations of dust concentrations.

I have furthermore an important remark on the discussion provided in Section 2.4 and on the use of {H2O,$\delta$D} pair remote sensing data. As shown in Schneider et al. (2016, and references therein) it is important to ensure that the H2O and $\delta$D remote sensing products represent the same vertical altitudes, otherwise defective interpretations of the {H2O,$\delta$D} pair distributions are very likely. For this purpose the MUSICA {H2O,$\delta$D} pairs are generated by an a posteriori processing (the Type 2 product, which ensures that the {H2O,$\delta$D} pair distributions can be correctly interpreted). I think it would be important to clarify how this problem has been addressed for the here presented {H2O,$\delta$D} pair data.

We are aware of the post processing methodology the referee proposed to ensure that $\delta$D and humidity vertical profiles are representative of the exact same vertical profile. We agree that such post processing simplifies the interpretation of $\delta$d-H2O pairs for quantitative purpose. However in this study, we did not apply the post processing method on our IASI retrievals. The reason for that is the data used in that study have been processed a long time ago (the a posteriori processing on IASI data is now operational) and after verification on some samples of our data, we found that applying the post processing does not affect significantly the interpretation of our results. Moreover it is fine from an optimal estimation point of view to do so (in that case the smoothing contribution to the error is greater for $\delta$D than for H2O and $\delta$D are less precise than q). We show an example of the post processing in Figure S1 for one day of data above the North Atlantic (0-40°N/40°W-5°E) for data at 3.5 km and data at 5.5 km. As one can see there are indeed changes. The changes are especially important for humid data at 3.5 km (remind that we do not use data at 3.5 km) within convective area which is because when degrading the humidity profile at $\delta$D resolution, the vertical resolution (of H2O) becomes larger and at this altitude, more sensitive to the boundary layer which is very humid within this area. At 5.5 km there are also differences but the latter are unlikely to affect our analysis in $\delta$D-q space. We should also be careful when translating the results of Schneider et al., 2016 to our retrievals. As you mentioned the retrieval schemes are different and are likely to be differently sensitive. For example, we indeed use a relatively small spectral range centered on HDO maximum Jacobians and avoid fitting a large region where there is a lot of information on H2O. Our retrieval is thus not optimized to provide high resolution vertical profile of H2O and their AVK are more similar to $\delta$D ones than in the MUSICA retrieval scheme. Nevertheless, while we believe the post-processing should not affect the interpretation of the present study, it is worth noting that the a posteriori processing has recently been adopted in the processing of IASI data for simplifying the analysis made by end-users.

[Figure]

*Figure 1 Differences between retrieved δD -q pairs at 3.5 and 5.5 km with and without a posteriori processing*

In the data section, we now specify that the sensitivity of Dd and q are not the same:

> It is also important to mention that the δD and humidity retrieved profiles are not exactly representative of the same atmosphere, the humidity profile having more vertical information than δD. It is thus important to keep in mind that when δD-q pairs are considered, the δD estimate is representative of a thicker layer of the atmosphere than the q estimate.

I think it would be also important to mention that the transport out of the Saharan boundary layer to the atmosphere above the Canary Archipelago has been studied since many years mainly by the aerosol community (there are leading experts at the Izaña Observatory) and there are a lot of publications available, which I would like to recommend considering (e.g. Rodriguez et al., 2011, http://www.atmoschem-phys.net/11/6663/2011/ or Rodríguez et al., 2015, http://www.atmos-chemphys.net/15/7471/2015/, and references therein). Also interesting in this context could be to have a look on the works published for a current ACP/AMT special issue (http://www.atmos-chemphys.net/special_issue382.html).

We feel that discussing the transport of aerosols towards the Izana Observatory is out of the scope of the study and for the sake of clarity and conciseness we have decided not to dedicate a specific paragraph on the aerosol transport.

As a minor remark I would like to recommend not talking about "data above Izaña" (Izaña is a hill on Tenerife Island and the name of an observatory on this hill). When talking about a region that covers actually the whole Canary Archipelago I would like to recommend using something like "data representative for the Canary Archipelago region". Finally, I would like to recommend considering for the Introduction Section a reference to the review of Galewski et al. (2016, Rev. Geophys., 54, doi:10.1002/2015RG000512).

We now speak of the Canary Archipelago region.

We now cite the review paper by Galewsky.

---

## Author Response (AR2)

Dear Heini,

Thank you very much for your quick feedback and for your additional comments.

1) The confusing sentence has been deleted, the beginning of the paragraph now reads:

    *''In this section, we assume that the monthly average isotopic composition above the CAR in summer is the result of mixing between upper tropospheric air and boundary layer air, the control of the ratio of the number of the western air-masses to the eastern airmasses can thus be understood in terms of the mixing fraction of the humid source (f in equations 2 and 3) in a mixing model.''*

2) We now added reference on the aerosol transport studied by the aerosol community in section 3.2:

    *"As the warm and dry air moves off the African coast, the SABL rises and becomes the Saharan Air Layer (SAL) undercut by the cool and moist marine boundary layer.* *The SAL contributes to important dust transport over the Atlantic (e.g. Prospero and Carlson 1972) and has been widely studied by the aerosol community (e.g. Ben-Ami et al. 2009; Rodríguez et al. 2011)."*

Best regards,

Jean-Lionel Lacour